

# Redescription of *Aspidosiphon* (*Paraspidosiphon*) *steenstrupii* Diesing, 1859 (Sipuncula: Aspidosiphonidae) and the reinstatement of three species

Itzahi Silva-Morales and Luis F. Carrera-Parra

Departamento de Sistemática y Ecología Acuática, El Colegio de la Frontera Sur, Chetumal, Quintana Roo, Mexico

## ABSTRACT

Sipuncula, specifically the family Aspidosiphonidae, faces taxonomic challenges due to brief original descriptions and the poor condition or loss of the type material. Detailed standardized redescriptions are essential to understanding the diversification in this group. Herein, a comprehensive redescription of *Aspidosiphon* (*Paraspidosiphon*) *steenstrupii* based on an extensive material collection from the tropical Western Atlantic is provided. Based on morphological data and the analysis of COI sequences, we delimited *A.* (*P.*) *steenstrupii* morphologically, restricting its distribution to the tropical Western Atlantic. Also, the redescriptions and proposals for reinstatement of *A.* (*P.*) *exostomum*, *A.* (*P.*) *ochrus*, and *A.* (*P.*) *speculator*, previously considered junior synonyms of *A.* (*P.*) *steenstrupii*, are included. Furthermore, a comprehensive discussion on diagnostic morphological features to recognize aspidosiphonid species and a detailed revision of synonyms of *A.* (*P.*) *steenstrupii* are included. Notable differences in morphology and genetic data suggest the need for revising the taxonomic status of several synonyms within the family, highlighting underestimated diversity in sipunculans.

## INTRODUCTION

Sipuncula is a group of unsegmented marine worms currently classified within the phylum Annelida (*Struck et al., 2007*; *Lewin, Liao & Luo, 2024*), with about 160 formally recognized species worldwide (*Schulze & Kawauchi, 2021*). Since 2021, ten additional species have been described (*Silva-Morales & Gómez-Vásquez, 2021*; *Dixit, Silva-Morales & Saravanane, 2023*; *Gómez-Vásquez, 2024*; *Hsueh & Glasby, 2024*; *Maiorova, Morozov & Adrianov, 2024*). Despite recent advances in revising the phylogeny and higher classification of the group (*Kawauchi, Sharma & Giribet, 2012*), many taxonomic issues remain to be addressed. Unfortunately, only a few specialists worldwide have focused on their study, particularly from the taxonomic and systematic perspective. Moreover, sipunculans exhibit relatively simple morphological features; they are soft, muscular, elongated worms with a general lack of external anatomical traits compared to other marine invertebrates, such as polychaetes or crustaceans (*Saiz-Salinas, 2018*). Consequently, the limited morphological characters

Corresponding author
Luis F. Carrera-Parra,
lcarrera@ecosur.mx

available hinder accurate species identification. In one of the six families of Sipuncula, the family Aspidosiphonidae *De Quatrefages, 1866*, the original species descriptions are often brief and frequently lack illustrations. This problem leads to possible taxonomic misidentifications, which could suggest that most, if not all, species have already been accounted for. Other problems are the poor condition of the type material, and the loss of some type specimens. Therefore, detailed and standardized morphological redescriptions of both type and non-type material are necessary. Searching for and reviewing topotype material is essential to resolving the aforementioned taxonomic issues and clarifying the taxonomic status of species.

The authorship of the family has been erroneously assigned to *Baird (1868)*; however, it was *De Quatrefages (1866)* who designated the family under the name Aspidosiphonea with *Aspidosiphon* Diesing, 1851 as type genus. *Baird (1868)* only corrected the name to the current spelling, Aspidosiphonidae. In accordance with article 11.7.1.3 of the International Code of Zoological Nomenclature (ICZN) which states that a family-group name of which the family-group name suffix is incorrect is available with its original authorship and date, but with a corrected suffix; for that reason, Aspidosiphonidae is attributed to *De Quatrefages (1866)*, not to the author who first corrected the spelling.

Since 1994, three new species of Aspidosiphonidae have been described, one from Thailand, *Aspidosiphon* (*Akrikoides*) *quadratoides* (*Hylleberg, 2014*) and two from Mexico, *Aspidosiphon* (*Paraspidosiphon*) *cutleri* and *A.* (*P.*) *pastori* (*Silva-Morales & Gómez-Vásquez, 2021*) bringing the total to 77 species, including synonymies and valid species. In the only worldwide revision of the family Aspidosiphonidae, the total number of species was reduced from 64 (*Stephen & Edmonds, 1972*) to 19 valid species (*Cutler & Cutler, 1989*); 70% of species names were synonymized.

The extensive synonymization was partially based on the assumption that morphological differences between geographically distant populations were insignificant or insufficient to separate the species (*Cutler & Cutler, 1989*), and the wide geographic distributions of sipunculans species were attributed to the supposed high dispersal capability of species with teleplanic pelagosphera larvae. These larvae were believed to undergo a pelagic development phase long enough to enable dispersal across the Atlantic Basin *via* major transoceanic currents (*Scheltema & Hall, 1975*). This idea was further supporter by laboratory experiments that showed this type of larva could remain in the water column for up to six months (*Rice, 1976*). However, recent molecular analyses and detailed morphological revisions have revealed cryptic and pseudocryptic species, indicating potential taxonomic problems at the species level in Sipuncula. These studies have rejected previously assumed wide distributions for some species (*Staton & Rice, 1999*; *Schulze et al., 2012*; *Kawauchi & Giribet, 2010*; *Kawauchi & Giribet, 2014*; *Johnson et al., 2016*; *Silva-Morales et al., 2019*; *Silva-Morales, 2020*).

In particular, *Aspidosiphon* (*Paraspidosiphon*) *steenstrupii Diesing, 1859* has seven synonyms and this species is considered to have a worldwide distribution "*throughout the western and northern Indian Ocean, from northern Australia through Indonesia, Vietnam, and the South China Sea, to southern tropical Japan and out through the western Pacific islands to Hawaii. Also collected from numerous Caribbean locations, in the eastern Atlantic*

*only from the Cape Verde Islands and the Gulf of Guinea. It lives in shallow-water coral rocks*" (*Cutler, 1994*).

Herein, a detailed morphological redescription of *A.* (*P.*) *steenstrupii*, as well as three species considered its junior synonyms (*A.* (*P.*) *exostomum Johnson, 1964*, *A.* (*P.*) *ochrus Cutler & Cutler, 1979*, and *A.* (*P.*) *speculator Selenka, 1885*), based on type and topotypic specimens, are provided. Furthermore, we discuss the taxonomic status of synonyms based on morphological data for all examined species and molecular data for two of them.

## MATERIALS & METHODS

Specimens from the following collections were reviewed: The British Museum of Natural History (BMNH), London, England; Colección de Bentos Costero (ECOSUR), El Colegio de la Frontera Sur, Chetumal, Mexico; Museum of Comparative Zoology (MCZ), Harvard University, Massachusetts, USA; National Museum of the Natural History (USNM), Smithsonian Institution, Washington, USA. Muséum National d'Histoire Naturelle (MNHN), Paris, France; National Museums of Scotland (NMS), Edinburgh, Scotland; Invertebrate Collections of the Florida Museum of Natural History (UF), University of Florida, USA; Marine Invertebrate Museum (UMML), Rosenstiel School of Marine and Atmospheric Science, University of Miami, Florida, USA.

The species redescriptions were primarily based on type materials, with comments on additional specimens to assess potential intraspecific variations. We did not provide a diagnosis for the species because, in groups with closely morphological boundaries between species, diagnoses can lead to confusion in recognizing new taxa or reinstating species from their synonyms, particularly if they rely solely on the few characters included in the diagnosis. Standardized descriptions included external and internal anatomy, following the terminology proposed by *Cutler & Cutler (1989)* and *Cutler (1994)*. Nevertheless, we propose the recognition of a new type of introvert hook. Traditionally, Aspidosiphonidae introvert hooks have been classified into three types: Type A: compressed hooks, Type B: pyramidal hooks, and Type C: conical hooks. Following our morphological revision, we identified a previously unrecognized type of hook. These hooks are distributed in the posterior region of the introvert and were previously classified under Type B (pyramidal hooks). We now propose designating them as Type D: leaf hooks.

Type D: leaf hooks differ from Type A: compressed hooks in that they lack lateral compression. In contrast to the triangular base characteristic of Type B: pyramidal hooks and the nearly circular base of Type C: conical hooks, Type D: leaf hooks feature an irregular base. Additionally, the tip and lateral profile of Type D hooks resemble the shape of a leaf, further distinguishing them from the other hook types.

Introvert hooks and trunk papillae were extracted with fine forceps and examined under a compound light microscope. Hooks were excised from three different regions (anterior, median, and posterior) of the introvert, while papillae were described from three different regions (anterior, median, and posterior) of the trunk. These structures were further analyzed using scanning electron microscopy (SEM) to obtain detailed observations. For SEM preparation, the entire introvert was dehydrated using a graded

series of hexamethyldisilazane (HMDS). After air drying, the introvert was mounted on an aluminum stub and coated with gold for imaging with a JEOL JSM-6010Plus-LA scanning electron microscope at the Scanning Electron Microscopy Laboratory (LMEB), ECOSUR-Chetumal. Digital images of selected internal and external features were captured using a Canon X6 digital camera mounted on a dissecting stereomicroscope. To enhance depth of field, all images were processed from a series of optical focal planes using HeliconFocus v6.7.1.

Eight sequences of the cytochrome c oxidase subunit 1 (COI) with an alignment length of 544 bp were used for molecular analyses. Sequences of *Aspidosiphon* (*Paraspidosiphon*) *steenstrupii* (DQ300119.1 from Barbados, DQ300116.1 from Florida, BCGG174-19 from Panama) and *A.* (*P.*) *exostomum* (as *A.* (*P.*) *steenstrupii* DQ300117.1 from Hawaii, DQ300118.1 from Thailand) were retrieved from GenBank. Additional sequences were included for comparison: one of *A.* (*P.*) *parvulus* Gerould, 1913 (DQ300115.1 from Belize), one of *Aspidosiphon* (*Aspidosiphon*) *gosnoldi* Cutler, 1981 (DQ300109.2 from Florida), one of *A.* (*A.*) *muelleri* Diesing, 1851 (DQ300113.2 from France), and one of *Cloeosiphon aspergillus* de Quatrefages, 1866 (DQ300120.1 from South Africa).

All sequences were aligned using the ClustalW method. The best substitution model was selected based on the lowest Bayesian Information Criterion (BIC). Based on the BIC results, the Tamura-Nei 1993 (TN93) model using a discrete Gamma distribution (+G) with five rate categories and by assuming that a certain fraction of sites is evolutionarily invariable (+I) was used to construct a tree by maximum likelihood analysis. Additionally, the Kimura 2-parameter model (*Kimura, 1980*) was used to estimate the average evolutionary divergence over sequence pairs within and between species. All analyses were carried out with Mega 11 (*Tamura, Stecher & Kumar, 2021*).

This published work and the nomenclatural acts it contains have been registered in ZooBank, the online registration system for the ICZN. The ZooBank LSIDs (Life Science Identifiers) can be resolved and the associated information viewed through any standard web browser by appending the LSID to the prefix http://zoobank.org/. The LSID for this publication is: urn:lsid:zoobank.org:pub:948402A8-B512-4102-B02E-0C0C1F7DC557. The online version of this work is archived and available from the following digital repositories: PeerJ, PubMed Central, and CLOCKSS.

# RESULTS

## Systematics

**Order Sipuncula Sedgwick, 1898**
**Family Aspidosiphonidae de Quatrefages, 1866**
**Genus *Aspidosiphon* Diesing, 1851**

**Type species.** *Aspidosiphon muelleri Diesing, 1859*, by subsequent designation.
**Diagnosis (after *Cutler, 1994*).** Introvert usually longer than trunk. Recurved hooks in numerous rings (absent in three species, scattered in two). Trunk with anal shield

composed of hardened units which may be inconspicuous. Introvert protrudes from ventral margin of shield. Body wall either with continuous longitudinal muscle layer or with longitudinal muscle layer separated into anastomosing, sometimes ill-defined bundles. Tentacles enclose dorsal nuchal organ but not mouth. Contractile vessel without villi. Two introvert retractor muscles may be almost completely fused. Spindle muscle attaches posteriorly. Two nephridia.

### Subgenus *Aspidosiphon* (*Paraspidosiphon*) Stephen, 1964

**Type species.** *Aspidosiphon steenstrupii* Diesing, 1859, by original designation.

**Diagnosis.** (after Cutler, 1994). Introvert with compressed hooks in rings, longitudinal muscle layer divided into bundles (LMB), most of them anastomosed.

### *Aspidosiphon* (*Paraspidosiphon*) *steenstrupii* Diesing, 1859, restricted.
Figures 1–2

*Aspidosiphon steenstrupii* Diesing, 1859:767–768, tab. 2, figs. 1–6. Migotto & Ditadi, 1988: 259–260, figs. 6A–6E.
*Aspidosiphon semperi* Ten Broeke, 1925:92, figs. 18–20.

**Type locality.** Saint Thomas, Virgin Islands, USA.

## Material examined.

**USA, *Florida*.** UMML, 1 specimen. Margot Fish Shoal, Dade Co, Apr 5, 1966, coll. G. Hendrix, bored in coral rubble. MCZ 130152, (GenBank DNA 100232 (DQ300116.1)), 1 specimen, Pickles Reef, Key Largo, Florida, USA, Nov 27, 1993, coll. S. Taylor. **Bahamas. *Near Ragged Island*.** UMML P-1442, 1 specimen, R/V Pillsbury, Cruise 7106, sta. 1442, 22°09′00″N, 75°35′00″W, 18 m, Jul 24, 1971. **Lesser Antilles. Saint Martin, *NE side of St. Martin.*** UF 331, 1 specimen, 18°07′48″N, 63°00′18″W, canyon with sponges, in algae, 13 m, Apr 11, 2012, coll. S. Rulliet. **Mexico, mexican Caribbean. *Cancun, Punta Nizuc*.** ECOSUR-S229, 2 specimens, 21°02′02.07″N, 86°46′41.20″W, coralline rock, 1.5 m, Feb 10, 2001, coll. S. Salazar-Vallejo. ***Cozumel*.** ECOSUR-S231, 1 specimen, in front to SEDENA, 20°31′00.61″N, 86°56′45.52″W, coralline rock, 1.5 m, Mar 24, 2001, colls. S. Salazar-Vallejo, M. Londoño-Mesa. ECOSUR-S235, 4 specimens, Playa Azul, 20°32′51.98″N, 86°55′46.45″W, coralline rock, one m, Mar 25, 2001, colls. S. Salazar-Vallejo, L. Carrera-Parra. ***Tulum*.** ECOSUR-S236, 6 specimens, Playa Aventuras, 20°21′47.20″N, 87°19′53.10″W, coralline rock, 1.5 m, Feb 17, 2001, colls. S. Salazar-Vallejo, L. Carrera-Parra. ***Mahahual.*** ECOSUR-S0210, 2 specimens, back reef, 18°42′31.30″N, 87°42′30.39″W, coralline rock, two m, Mar 22, 2000, colls. S. Salazar-Vallejo, L. Carrera-Parra. ECOSUR-S0211, 1 specimen, 50 m off coast, 18°43′38.68″N, 87°41′56.81″W, coralline rock, two m, Mar 4, 1998, colls. S. Salazar-Vallejo, L. Carrera-Parra. ECOSUR-S218, 1 specimen, reef lagoon, 18°43′24.93″N, 87°42′02.95″W, coralline rock, one m, Jan

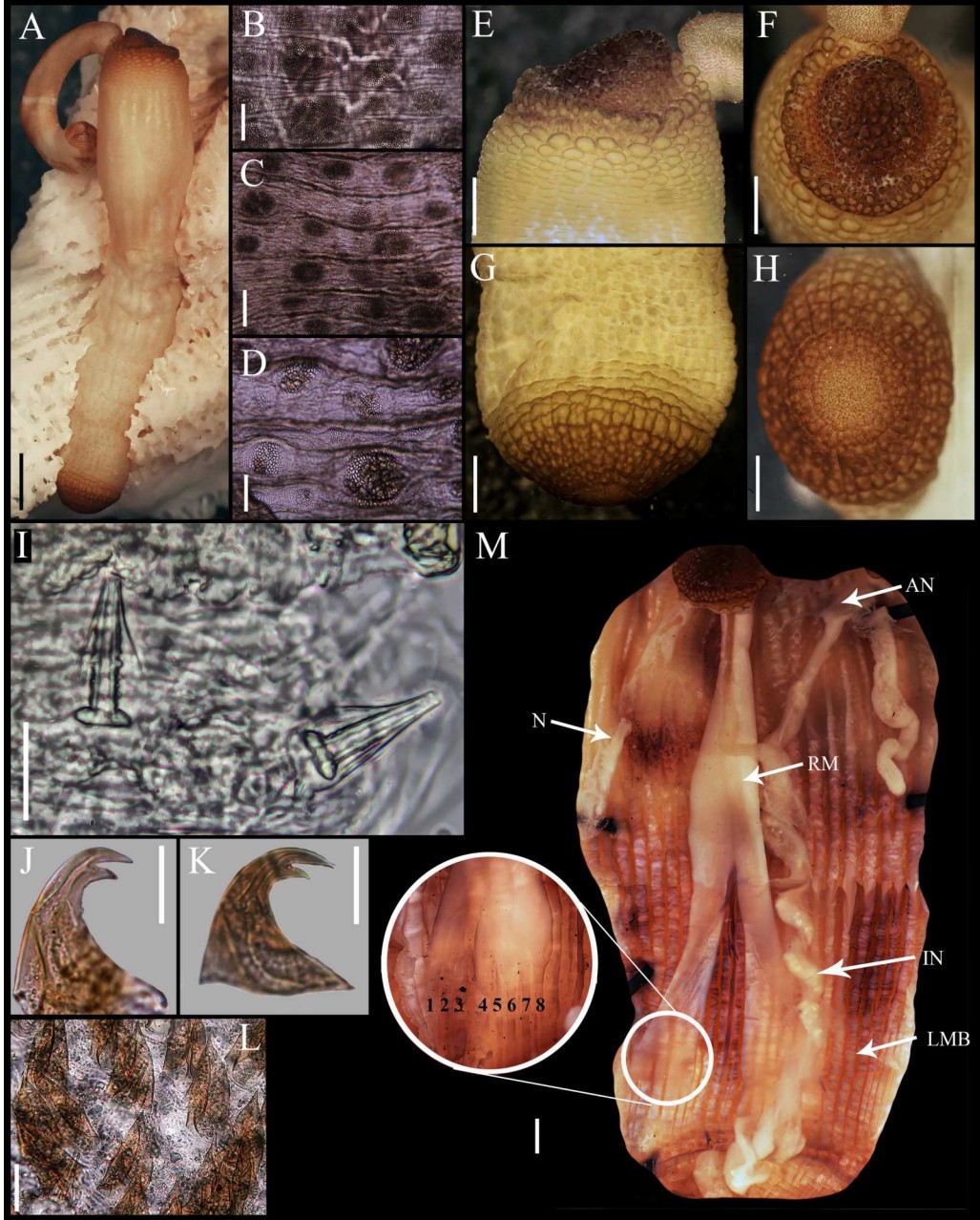

**Figure 1** *Aspidosiphon* (*Paraspidosiphon*) *steenstrupii* *Diesing, 1859*. **UF331, Saint Martin, Lesser Antilles.** (A) Adult body plan, lateral view. (B) Papillae from anterior region of trunk. (C) Papillae from median region of trunk. (D) Papillae from posterior region of trunk. (E) Anal shield, lateral view. (F) Anal shield, dorsal view. (G) Caudal shield, lateral view. (H) Caudal shield, frontal view. (I) Papillae from median region of introvert. (J) Hook from posterior ring. (K) Hook from anterior rings. (L) Leaf hooks. (M) Internal morphology of a dissected specimen. Abbreviations. AN, anus; IN, intestine; LMB, longitudinal musculature bundles; N, nephridia; RM, retractor muscles. Scale bars. A: two mm, B–D: 0.1 mm, E–H: 0.05 mm, I–L: 25 μm, M: one mm.

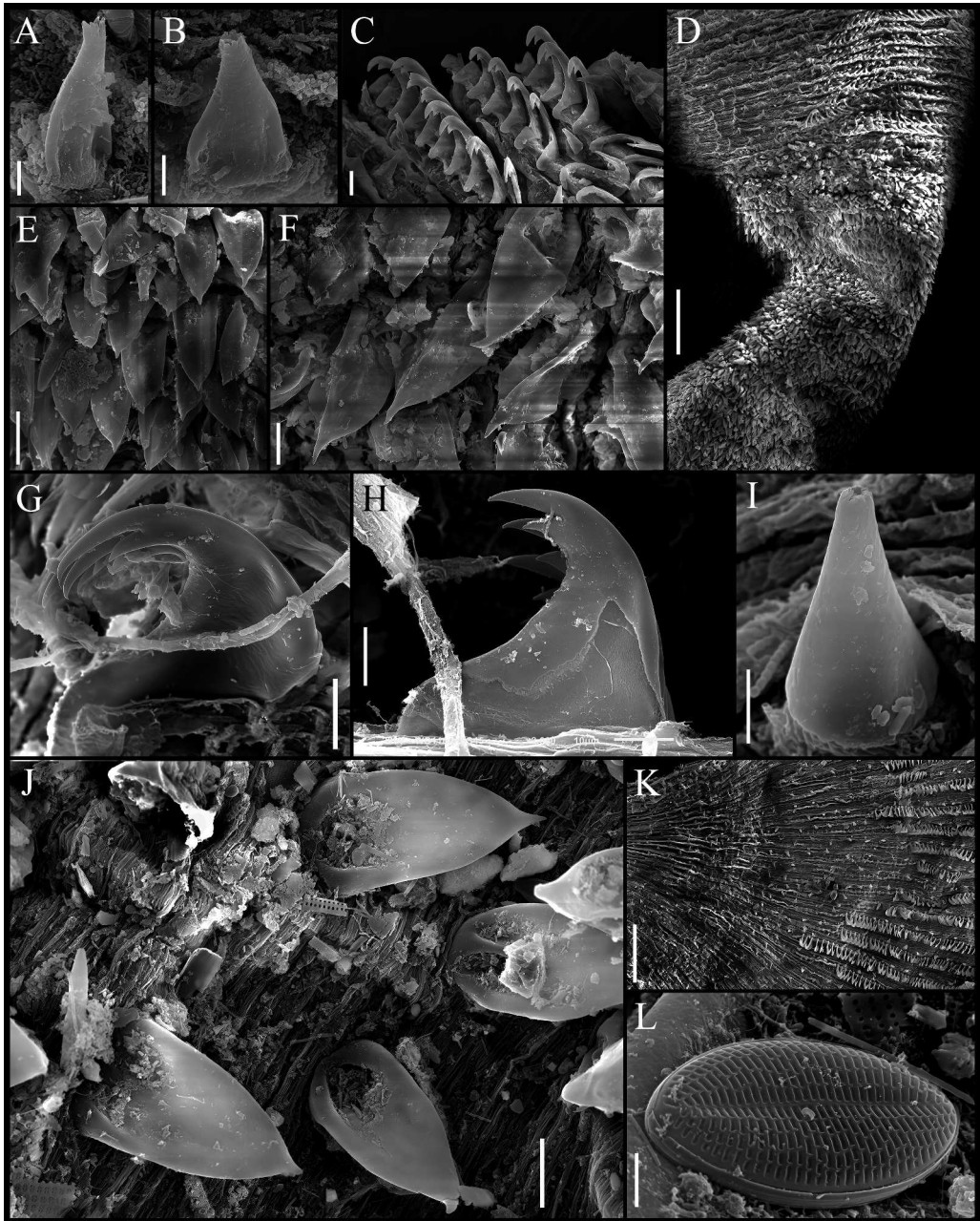

**Figure 2** *Aspidosiphon* (*Paraspidosiphon*) *steenstrupii Diesing, 1859*. **SEM images. UF331 Saint Martin, West Indies.** (A) Conical papillae from anterior introvert. (B) Conical papillae from median introvert. (C) Hooks from first rings. (D) Transition zone between rings and dispersed hooks. (E) Leaf hooks from anterior region. (F) Leaf hooks from posterior region. ECOSUR-S236, Tulum, Mexico. (G) Hooks from first rings. (H) Hooks from posterior rings. (I) Conical papillae from median region of the introvert. (J) Leaf-pyramidal hooks from anterior region. (K) Transition zone between ventral and dorsal introvert. (L) Diatom found in introvert. Scale bars. A–B, I: five μm, C, E: 20 μm, D, K: 200 μm, F–H, J: 10 μm, L: two μm.

19, 2001, colls. P. Salazar-Silva, J. Bastida-Zavala, M. Tovar-Hernández, S. Salazar-Vallejo, L. Carrera-Parra. ECOSUR-S220, 1 specimen, fore reef, 18°42′43.32″N, 87°42′22.51″W, coralline rock, 15 m, Jun 6, 1998, coll. M. Ruiz-Zárate. ECOSUR-S222, 1 specimen, 50 m off coast, 18°43′38.68″N, 87°41′56.81″W, coralline rock, two m, Mar 4, 1998, colls. S. Salazar-Vallejo, L. Carrera-Parra. ECOSUR-S223, 1 specimen, reef lagoon near to back reef, 18°42′34.01″N, 87°42′31.22″W, coralline rock, 1.5 m, Jan 9, 2001, colls. P. Salazar-Silva, J. Bastida-Zavala, M. Tovar-Hernández, S. Salazar-Vallejo, L. Carrera-Parra. ECOSUR-S225, 1 specimen, reef lagoon, 18°42′34.66″N, 87°42′32.27″W, coralline rock, 1.5 m, Mar 21, 2000, colls. J. Bastida-Zavala, P. Salazar-Silva. ECOSUR-S226, 2 specimens, without coordinates, with sponges, Jul 21, 1982. ECOSUR-S227, 1 specimen, old wooden pier, 18°42′41.95″N, 87°42′35.98″W, fouling, one m, Feb 24, 2001, colls. P. Salazar-Silva, J. Bastida-Zavala, M. Tovar-Hernández, S. Salazar-Vallejo, L. Carrera-Parra, L. Harris. ECOSUR-S230, 1 specimen, reef lagoon, 18°43′21.01″N, 87°42′4.28″W, coralline rock, 1.5 m, Feb 24, 2001, colls. P. Salazar-Silva, J. Bastida-Zavala, M. Tovar-Hernández, S. Salazar-Vallejo, L. Carrera-Parra, L. Harris.ECOSUR-S234, 2 specimens, old wooden pier, 18°42′41.95″N, 87°42′35.98″W, fouling, one m, Mar 18, 2001, colls. P. Salazar-Silva, J. Bastida-Zavala, M. Tovar-Hernández, S. Salazar-Vallejo, L. Carrera-Parra. *Punta Herradura.* ECOSUR-S213, 2 specimens, 18°32′25.24″N, 87°44′28.28″W, coralline rock, four m, Oct 28, 1997, coll. S. Salazar-Vallejo, L. Carrera-Parra. ECOSUR-S233, 3 specimens, 18°32′25″N, 87°44′30″W, coralline rock, two m, Sep 28, 1996, coll. S. Salazar-Vallejo, L. Carrera-Parra. *Xahuayxol*. ECOSUR-S0212, 8 specimens, reef lagoon, 18°30′12.46″N, 87°45′29.79″W, coralline rock, two m, Jun 1 1997, colls. S. Salazar-Vallejo, L. Carrera-Parra. ECOSUR-S214, 6 specimens, 120 m off coast, 18°30′41.34″N, 87°45′24.63″W, coralline rock, 1.5 m, Oct 31, 1997, colls. L. Carrera-Parra, S. Salazar-Vallejo. ECOSUR-S215, 9 specimens, 18°32′25″N, 87°44′30″W, coralline rock, two m, Sep 28, 1996, coll. S. Salazar-Vallejo, L. Carrera-Parra. ECOSUR-S216, 5 specimens, reef lagoon, 18°30′39.77″N, 87°45′24.80″W, coralline rock, 1.8 m, Jun 4, 1998, colls. S. Salazar-Vallejo, L. Carrera-Parra. ECOSUR-S217, 1 specimen, 80 m off coast, 18°30′42.00″N, 87°45′25.80″W, sediment with sea grass, 1.73 m, Jun 1, 1997, colls. L. Carrera-Parra, S. Salazar-Vallejo. ECOSUR-S219, 20 specimens, reef lagoon, 18°30′13.02″N, 87°45′32.12″W, coralline rock, 0.95 m, Sep 26, 1996, colls. S. Salazar-Vallejo, L. Carrera-Parra. ECOSUR-S221, 3 specimens, reef lagoon, 18°30′13.71″N, 87°45′31.50″W, coralline rock, one m, Jun 2 1998, colls. S. Salazar-Vallejo, L. Carrera-Parra. ECOSUR-S224, 8 specimens, 100 m off coast, 18°30′15.08″N, 87°45′30.98″W, in sediment with sea grass, two m, Sep 27, 1996, coll. S. Salazar-Vallejo, L. Carrera-Parra. ECOSUR-S228, 6 specimens, reef lagoon, 18°30′13.02″N, 87°45′32.12″W, coralline rock, 0.95 m, Sep 26, 1996, colls. S. Salazar-Vallejo, L. Carrera-Parra. ECOSUR-S232, 2 specimens, reef lagoon, 18°30′12.46″N, 87°45′29.79″W, coralline rock, two m, Jun 1 1997, colls. L. Carrera-Parra, S. Salazar-Vallejo. ECOSUR-S237, 13 specimens, reef lagoon, 18°30′39.04″N, 87°45′25.09″W, coralline rock, 1.7 m, Sep 27, 1996, colls. L. Carrera-Parra, S. Salazar-Vallejo. **Dominican Republic.** UMML P-1272, 1 specimen, R/V Pillsbury, Cruise 7006, sta. 1272, off Cabo Rojo, 17°52′41.98″N, 71°41′12.01″W, 20-27 m, Jul 17, 1970, coll. J. Staiger. **Panama.** *Bocas del Toro.* UF495, [BCGG|174-19], 1 specimen, 9°20′31.20″N, 82°15′36″W, May 23, 2016, colls. M. Leray, F. Michonneau, R. Lasley.

(Identified as *Aspidosiphon* (*P.*) *laevis*). **Barbados.** MCZ 130155, (GenBank DNA 100630 (DQ300119.1)), 1 specimen, Barbados, Bank Reef, 13°11′21.8″N, 59°34′33.3″W, Jun 26, 2002, coll. J.I. Saiz-Salinas, A. Schulze, Id. A. Schulze.

### Redescription.

Male specimen from Saint Martin, West Indies (UF331).

External morphology. Trunk 16 mm in length (Fig. 1A); smooth, white with opaque body wall; trunk with semicircular papillae with platelets, different sizes, non-conglomerate, (30–100 μm length), arranged scattered throughout the trunk (Figs. 1B–1D).

Introvert almost entirely protruded, 50% the trunk length. Introvert papillae conical (Fig. 1I), shorter anteriorly (36–38 μm), taller posteriorly (40–42 μm), arranged in rings, each papilla with about six denticles (Figs. 2A–2B). Tentacles not observed.

Introvert with the most anterior hooks (60–70 μm tall, $n = 20$ rings, $n = 240$ hooks) Type A, (compressed, bidentate) arranged in more than 50 rings (Figs. 1J–1K, 2C–2D, 2G–2H). Hooks from first rings with the angle between line X and Y more than 90° (Figs. 2C, 2G), clear streak with ill-defined basal triangle and well-defined tongue-like extension; hooks from posterior rings with well-defined basal triangle and well-defined tongue-like extension (Figs. 1J, 2H). Hooks all Type A with main and secondary tooth, sharp, main tooth does not exceed the length of the hook base. Hooks Type A followed by Type D, dark leaf hooks (40–50 μm tall), arranged dispersedly (Figs. 1L, 2D–2F, 2J).

Anal shield dark brown, thick, margins ill-defined without grooves (Figs. 1E–1F). Conical units in the center of shield, and conglomerate spherical units marginally. Shield with crater-like form with a conical protuberance (one mm tall) in lateral view, protuberance vertex oriented toward the introvert.

Caudal shield dark brown with shallow grooves arranged radially, margins ill-defined, semi-rectangular units in the margin (Figs. 1G–1H).

Internal morphology (Fig. 1M). A pair of nephridia opening at the anus level, attached along 25% of their length, occupying 55% trunk length. Longitudinal muscle layer divided into 18 anastomosed LMB in the anterior region of trunk, 24 in the median region, and 30 in the posterior region. A pair of retractor muscles attached to the body wall in the 80% of the trunk length, each retractor muscle attached to 8 bundles, starting from the second bundle after the ventral nerve cord, fused on the half of their length. Wing muscle and caecum present. Eyespots not observed. Fixing muscle attached in the median region of the trunk, splitting into two before the rectum and attached anterior to the anus. Spindle muscle attached posteriorly.

**Habitat.** In coralline rock and fouling; 1–27 m depth.

**Distribution.** From Florida to Brazil. Restricted to the tropical Western Atlantic; other records are questionable.

**Variations.** Trunk length (8–25 mm, $n = 105$). Nephridia length (55–82% trunk length, $n = 105$). Retractor muscles attachment (77–82% trunk length, $n = 50$). Longitudinal muscle bundles in the median region (24–26 LMB, $n = 50$).

**Remarks.** The type material is lost; therefore, the redescription is based on a specimen from Saint Martin, the closest locality to Saint Thomas, the type locality of *A.* (*P.*) *steenstrupii*.

Additionally, we examined specimens from several localities from Florida to Barbados, including a large set of specimens collected along the Mexican Caribbean.

When *Diesing (1859)* described *Aspidosiphon* (*P.*) *steenstrupii*, he did not include a detailed description of the papillae and hooks. In this contribution, we present SEM photographs of the papillae and hooks for the first time, providing a detailed description of these characters. However, it is crucial to emphasize that diagnostic characters can be observed under a light microscope, and SEM is not essential for identifying sipunculan species.

The most evident morphological difference between our description and the original description by *Diesing (1859)* is seen in the skin of the trunk; Diesing described his specimen (trunk length of nine mm) with the anterior region of the trunk having longitudinal divisions and the posterior region with transverse divisions; however, in the specimen described here (trunk length of 16 mm) we only observed longitudinal divisions. After the revision of several specimens, we consider that this variation likely arises from longitudinal division in the skin, which may be attributable to trunk contraction in the specimens we examined.

When *Ten Broeke (1925)* described *Aspidosiphon semperi* from Caracas Bay, Curaçao, he indicated that the only difference between his new species and *A.* (*P.*) *steenstrupii* was the presence of four retractor muscles in *A. semperi* instead of two in *A.* (*P.*) *steenstrupii*. However, in subsequent studies, *Gibbs & Cutler (1987)* and *Cutler & Cutler (1989)* examined material from Curaçao and concluded there were only two retractor muscles. We agree with *Cutler & Cutler*'s (*1989*) proposal to synonymize *A. semperi* with *A. (P.) steenstrupii* because no species in Aspidosiphonidae has four retractor muscles, and therefore, this was most likely an observational error.

In the case of *Aspidosiphon trinidensis* described by *Cordero & Mello-Leitão (1952)* from Ilha da Trindade, Brazil, the description was based on a single specimen. The most important morphological difference between this species and *A.* (*P.*) *steenstrupii* is the absence of bidentate hooks in *A. trinidensis*. According to the authors, *A. trinidensis* differ by the number of LMB, the morphology of the hooks, the number of intestinal turns, and the shape and arrangement of the nephridia. However, they did not compare this species with *A. (P.) steenstrupii*, but with the "Kluzingeri-pachydermatus group" (=*A. (P.) laevis*).

*Cutler & Cutler (1980)* recorded *A. trinidensis* from the Bahamas but, this locality is far from the type locality of *A. trinidensis*. Later, when *Cutler & Cutler (1989)* carried out the revision of *Aspidosiphon*, they were unable to locate the type material of *A. trinidensis*. Upon re-examining their own material from the Bahamas, they found that it had bidentate hooks and concluded that the morphology of *A. trinidensis* was within the morphological variation of *A. (P.) steenstrupii*, leading them to synonymize *A. trinidensis* with *A. (P.) steenstrupii*.

We consider it important to note that the only specimen used to describe *A. trinidensis* had an invaginated introvert, making thorough study impossible. The absence of bidentate hooks, typically found in rings in the most anterior region of the introvert, likely led to the conclusion that this was a new species. We believe that this species is very similar to *A. (P.) steenstrupii*, but until the revision of the topotype material, we cannot make a definitive conclusion about the taxonomic status of *A. trinidensis*.

Sato (1939) described *Aspidosiphon makoensis* based on three specimens collected from dead coral-rock inMako, Taiwan (20–25 mm trunk length). Unfortunately, we cannot examine the type material because it is lost (Cutler & Cutler, 1981). Sato (1939) mentioned that this new species seems to be very closely allied to *A.* (*P.*) *steenstrupii*, and these two species may be distinguished from each other by different features of skin-papillae distributed on the body surface and the attachment of the retractor muscles. However, he did not describe the details of these morphological differences. Cutler & Cutler (1981) synonymized *A. makoensis* with *A.* (*P.*) *steenstrupii* on the grounds that the range of variations of the hooks, the clear streak and the retractor muscles overlap. However, no redescriptions were made by locality, but a general description with specimens from all over the world was provided

After our redescription of *A.* (*P.*) *steenstrupii* and based on Sato's drawings, we noticed that the morphology of the trunk papillae is similar: both species have semicircular papillae, with platelets and non-conglomerate; the retractor muscles are attached not from the caudal extreme, but close to it. However, Sato's specimens lack a tongue-like extension on the hook and the anal shield is not semi-conical in lateral view. Given that hooks are a significant diagnostic character, the reinstatement of this species should be considered after a thorough examination of topotypic material.

Hsueh, Cheng & Kou (2006) reported several aspidosiphonid specimens from Taiwan, including *A.* (*P.*) *steenstrupii*. These specimens could be used as topotype of *A. makoensis* because they were collected from its type locality. Unfortunately, the description provided is not detailed enough. The specimens were deposited at the National Museum of Natural Science (MNNS), Taichung, Taiwan, Republic of China; but their current availability for study could not be confirmed.

**Aspidosiphon (Paraspidosiphon) exostomum Johnson, 1964 reinst. stat.**
Figure 3

*Aspidosiphon exostomum* Johnson, 1964: 331–332, pl. 7, figs. 1–9.
*Paraspidosiphon exostomus.*—Stephen & Edmonds, 1972:244.

**Type locality.** Port Blair, Andaman Island.
**Material examined. Type** NMS Z.1965.32.2, 1 specimen, Port Blair, Andaman Island.
**Additional materials. Thailand**, *Phuket*. MCZ 130154, (GenBank DNA 100391 (DQ300118.1)), 1 specimen, Jan 31, 2001, coll. J. Hylleberg. **Hawaii, *Honolulu, Kewalo Reef*.** MCZ 130153, (GenBank DNA 100372 (DQ300117.1)), 1 specimen, Jan 25, 2001, coll. J. Bailey-Brock.

### Redescription.
Type NMS Z.1965.32.2
External morphology. Trunk 26 mm in length (Fig. 3A); rough, brown with opaque body wall; trunk with semi rectangular papillae with platelets, different sizes, non-conglomerate,

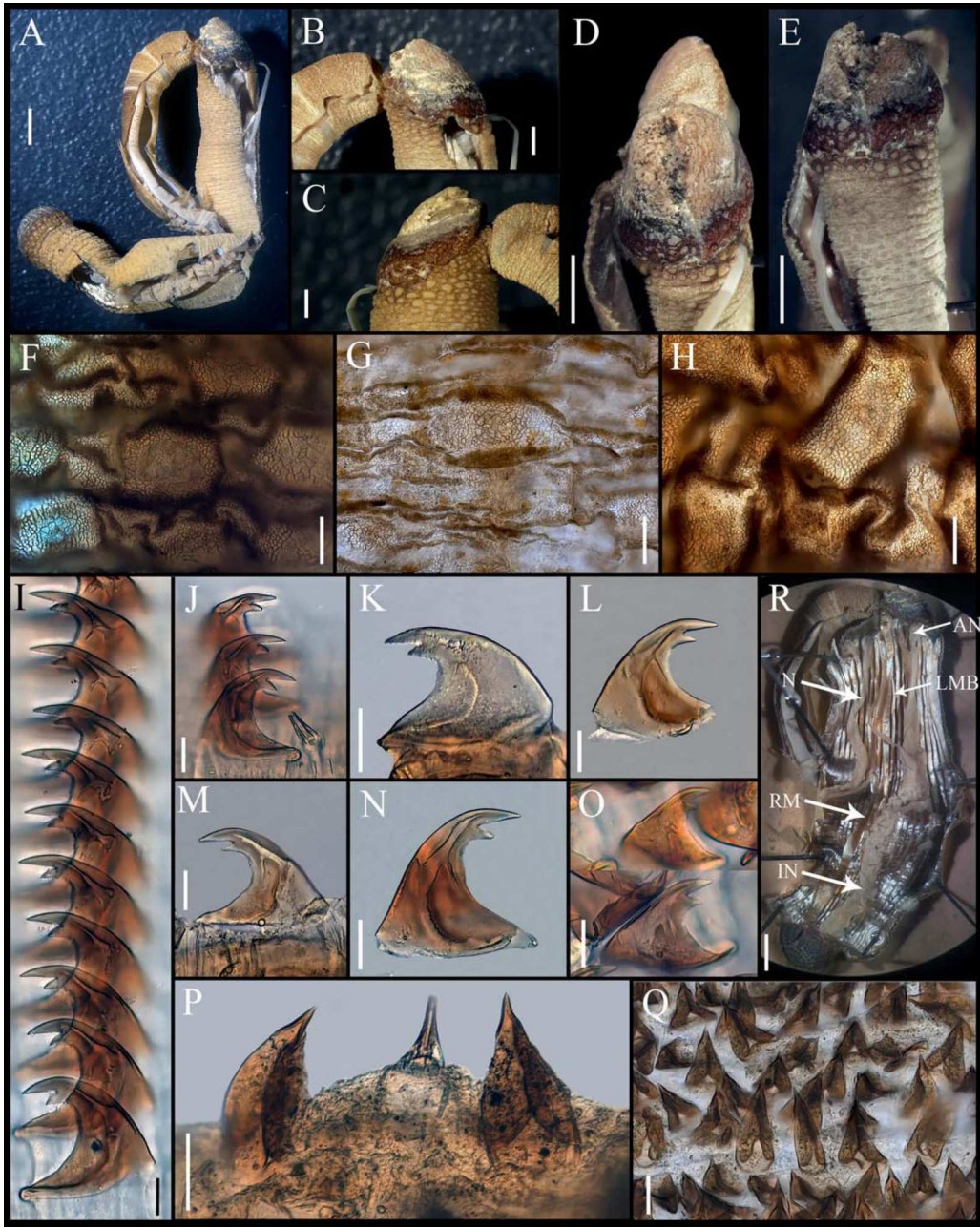

**Figure 3** *Aspidosiphon* (*Paraspidosiphon*) *exostomum Johnson, 1964*. **Type NMS Z.1965.32.2, Andaman Island.** (A) Adult body plan, lateral view. (B–C) Anal shield, lateral view. (D–E) Anal shield, dorsal view. (F) Papillae from anterior region of trunk. (G) Papillae from median region of trunk. (H) Papillae from posterior region of trunk. (I–N) Hooks from anterior rings. (O) Hooks from posterior rings. (P–Q) Leaf hooks. (R) Internal morphology. Abbreviations. AN, anus; IN, intestine; LMB, longitudinal musculature bands or bundles; N, nephridia; RM, retractor muscles. Scale bars. A, R: two mm, B–E: one mm, F–H: 100 μm, I–O: 25 μm, P–Q: 30 μm.

but close each other, (30–100 µm length), arranged scattered throughout the trunk (Figs. 3F–3H).

Introvert almost entirely protruded, 50% the trunk length. Introvert papillae conical (Figs. 3J, 3P) shorter anteriorly, taller posteriorly (25–30 µm tall), arranged in rings, each papilla with denticles. Tentacles not observed.

Introvert with the most anterior hooks (40–50 µm tall, $n = 8$ rings, $n = 82$ hooks) Type A, (compressed, bidentate), arranged in rings (Figs. 3I–3O). Hooks from anterior rings with the angle between line X and Y more than 90°; clear streak without basal triangle but with a large area of clear streak in the hook base, ill-defined tongue-like extension (Fig. 3I); hooks from posterior rings without basal triangle but with a large area of clear streak in the hook base, they can have 2–3 tongue-like extensions (Fig. 3O). Hooks all Type A with main and secondary tooth, sharp, main tooth does not exceed the length of the hook base. Hooks Type A followed by Type D, dark leaf hooks (50–60 µm tall), arranged scattered (Figs. 3P–3Q).

Anal shield dark brown, thick, margins ill-defined without grooves (Figs. 3B–3E). Conglomerate spherical units in the margins. Shield with crater-like form with an ill-defined conical protuberance.

Caudal shield dark brown with shallow grooves arranged radially, margins ill-defined, semi-rectangular units in the anterior margin (Fig. 3A).

Internal morphology (Fig. 3R). A pair of nephridia opening at the anus level, occupying almost 75% trunk length. Longitudinal muscle layer divided into 28–30 anastomosed LMB in the median region of the trunk. A pair of retractor muscles attached to the body wall in the 80% of the trunk length, each retractor muscle attached to 8 bundles, starting from the second bundle after the ventral nerve cord, fused on the half of their length. Fixing muscle present. Caecum and eyespots not observed. Spindle muscle attached posteriorly.

**Habitat.** Intertidal to 0.5 m depth in coralline rock.

**Distribution.** Andaman Island, Thailand, and Hawaii.

**Variations.** Trunk length (20–26 mm, $n = 3$). Nephridia length (55–75% trunk length, $n = 3$). Retractor muscles attachment (75–80%, $n = 3$). Longitudinal muscle bundles (28–33 LMB, $n = 3$).

**Remarks.** *Johnson (1964)* described *Aspidosiphon* (*Paraspidosiphon*) *exostomum* based on three specimens collected at Port Blair, Andaman Island. The type specimen measured 30 mm in length, including the height of the anal shield. The author indicated that the type specimens were deposited in the Museum of the Zoology Department at Birla College, Pilani, India. However, the specimen described here is now deposited in the National Museums of Scotland (NMS), and two slides (NMS Z.1965.32.2.1, Z.1965.32.2.2) were made and mounted in Euparal to preserve papillae and hooks.

Johnson argued that *A.* (*P.*) *exostomum* was distinct because its tentacles "form a crown dorsal to the mouth". *Cutler & Cutler (1989)* considered that this feature is shared among most species of Aspidosiphonidae (except for species that lack tentacles) and synonymized the species with *A.* (*P.*) *steenstrupii*. We agree that this feature does not provide taxonomic information at the species level; however, we do not agree about the synonymy because this is not the only difference we found.

After reviewing this material, we conclude that the name must be reinstated due to relevant morphological differences between *A.* (*P.*) *exostomum* and *A.* (*P.*) *steenstrupii*. The spherical papillae are distributed over a greater percentage of the body wall in *A.* (*P.*) *exostomum* compared to *A.* (*P.*) *steenstrupii*. The clear streak of the hooks in *A.* (*P.*) *exostomum* has a very broad region, covering more than half the width of the hook in contrast to the triangular region of the clear streak in *A.* (*P.*) *steenstrupii*, which is less than half the width of the hooks and has a clearer basal. The tongue-like extension in most posterior hooks arranged in rings of *A.* (*P.*) *exostomum* can vary from 1 to 3, whereas in *A.* (*P.*) *steenstrupii*, no more than one has been found. Furthermore, we noticed that in *A.* (*P.*) *exostomum* the anal shield does not have the conspicuous crater shape seen in *A.* (*P.*) *steenstrupii*. Finally, although both species have platelets on the trunk papillae, they exhibit different patterns of distribution: *A.* (*P.*) *exostomum* has more conglomerated papillae than *A.* (*P.*) *steenstrupii*. However, this difference should be considered carefully, as it may change depending on the degree of trunk contraction.

*Aspidosiphon* (*P.*) *exostomum* is a species with a wide distribution in the tropical Western Pacific, including Hawaii. We found no morphological differences between specimens examined, even in genetic data (see below). This type of wide distribution in this region has been previously reported in other organisms, such as the annelids *Hesione paulayi* Salazar-Vallejo, 2018; Salazar-Vallejo, 2018 and *Iphione picta* Kinberg, 1856 (Piotrowski et al., 2024).

### *Aspidosiphon* (*Paraspidosiphon*) *ochrus* Cutler & Cutler, 1979 reinst. stat.
Figures 4–5

*Aspidosiphon* (*Paraspidosiphon*) *ochrus* Cutler & Cutler, 1979: 976–978, figs. 15–17.

**Type locality.** Madagascar.
**Material examined. Papua New Guinea, *Bougainville Island*. Paratype** MNHN AH-405, 1 specimen, R/V Te Vega, Sta. 45-4, 6°12′S, 155°37′E, 32 m, Sep 10, 1963, Id. E.B. Cutler. **Additional materials. Australia, *Cocos-Keling Islands*.** USNM 64581, 2 specimens, R/V Te Vega, Sta. B-5, 12°00′S, 96°50′E, lagoon, one m, Jan 24, 1963.

### Redescription.
Notes on paratype MNHN AH-405.

External morphology. Trunk 10 mm in length (Fig. 4A); rough, brown with opaque body wall.

Introvert entirely protruded, 55% the trunk length. Tentacles digitiform (Figs. 4F–4G).

Anal shield dark brown, thick, margins ill-defined without grooves (Figs. 4B–4E). Conglomerate spherical units in the margins. Shield with flat form in lateral view.

Caudal shield dark brown with shallow grooves arranged radially, margins ill-defined, semi-rectangular units in the anterior margin.

Internal morphology (Fig. 4H). A pair of nephridia opening at the anus level, occupying more than 75% trunk length. Longitudinal muscle layer divided into 26 anastomosed

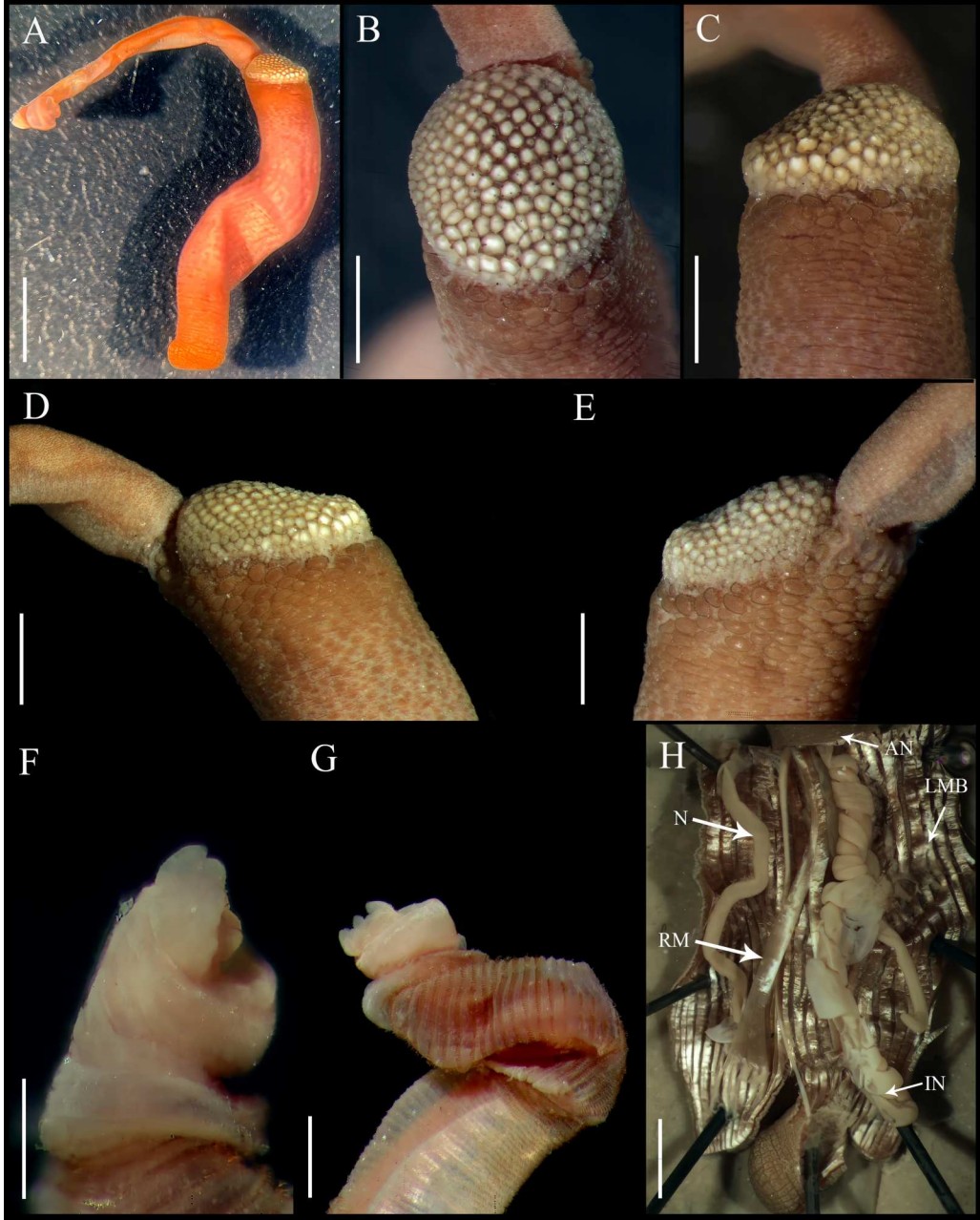

**Figure 4** *Aspidosiphon* (*Paraspidosiphon*) *ochrus Cutler & Cutler, 1979*. **Paratype MNHN AH-405, Bougainville Island, Papua New Guinea.** (A) Adult body plan, lateral view. (B–C) Anal shield, dorsal view. (D–E) Anal shield, lateral view. (F–G) Tentacles. (H) Internal morphology. Abbreviations. AN, anus; IN, intestine; LMB, longitudinal musculature bands or bundles; N, nephridia; RM, retractor muscles. Scale bars. A: three mm, B–G: one mm, H: two mm.

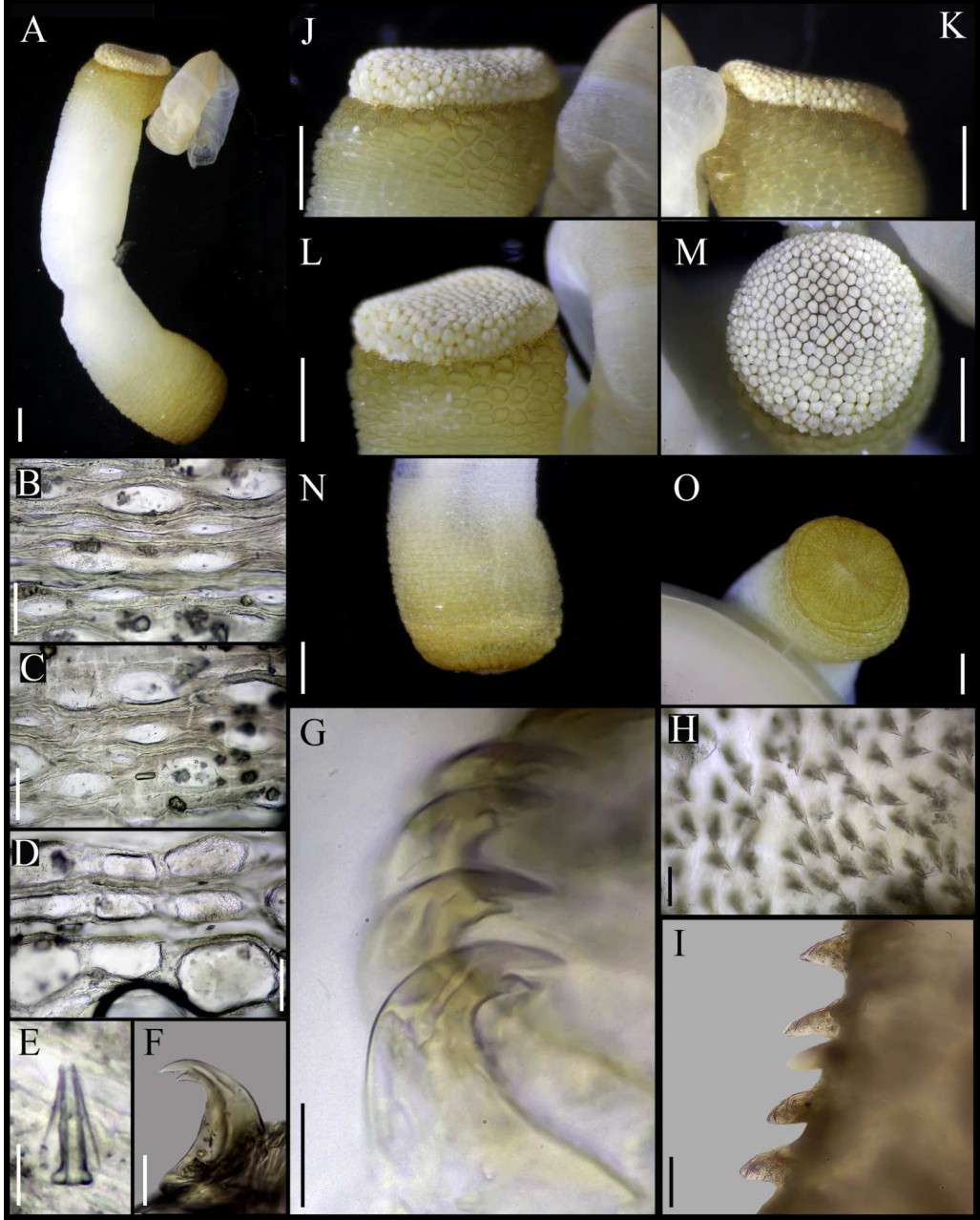

**Figure 5** *Aspidosiphon* (*Paraspidosiphon*) *ochrus* *Cutler & Cutler, 1979*. **Non-type specimen USNM 64581, Indian Ocean.** (A) Adult body plan, lateral view. (B) Papillae from anterior region of trunk. (C) Papillae from median region of trunk. (D) Papillae from posterior region of trunk. (E) Conical papillae from median introvert. (F) Hooks from anterior rings. (G) Hooks from posterior rings. (H) Leaf hooks. (I) Leaf hooks, lateral view. (J–K) Anal shield, lateral view. (L–M) Anal shield, dorsal view. (N) Caudal shield, lateral view. (O) Caudal shield, frontal view. Scale bars. A, N: two mm, B–D: 50 µm, E–G: 10 µm, H–I: 30 µm J–M, O: one mm.

LMB in the median region of the trunk, anastomosed. A pair of retractor muscles attached to the body wall in the 80% of the trunk length, each retractor muscle attached to five bundles, starting from the second bundle after the ventral nerve cord, fused on the half of their length. Fixing muscle, caecum and eyespots not observed. Spindle muscle attached posteriorly.

Specimen from USNM 64581

External morphology. Trunk 12 mm in length (Fig. 5A); smooth, white with opaque body wall; trunk with semicircular papillae with platelets, different sizes, non-conglomerate, (30–100 μm length), arranged scattered throughout the trunk (Figs. 5B–5D).

Introvert almost entirely protruded, 50% the trunk length. Introvert papillae conical (Fig. 5E), shorter anteriorly (20–23 μm), taller posteriorly (20–25 μm), arranged in rings, each papilla with denticles. Tentacles not observed.

Introvert with the most anterior hooks (25–30 μm tall, $n = 8$ rings, $n = 50$ hooks) Type A, (compressed, bidentate), arranged in rings (Figs. 5F–5G). Hooks from anterior rings with the angle between line X and Y more than 90°; clear streak without basal triangle but with a big area of clear streak in the hooks base, without tongue-like extension; hooks from posterior rings without basal triangle but with a big area of clear streak in the hook base, without tongue-like extension. Hooks all Type A with main and secondary tooth, sharp, main tooth does not exceed the length of the hook base. Hooks Type A followed by Type D, dark leaf hooks (20–25 μm tall), arranged scattered (Figs. 5H–5I).

Anal shield white, thick, margins ill-defined without grooves (Figs. 5J–5M). Units of similar size. Conglomerate spherical units in the margins. Shield with flat form in lateral view.

Caudal shield brown with shallow grooves arranged radially, margins ill-defined (Figs. 5N–5O).

Internal morphology. A pair of nephridia opening at the anus level, occupying 55% trunk length. Longitudinal muscle layer divided into 20 anastomosed LMB in the median region of the trunk. A pair of retractor muscles attached to the body wall in the 80% of the trunk length. Fixing muscle present. Caecum and eyespots not observed. Spindle muscle attached posteriorly.

**Habitat.** Coralline rock, 1–32 m.

**Distribution.** Indian Ocean.

**Variations.** Trunk length (10–12 mm, $n = 3$). Nephridia length (55–75% trunk length, $n = 3$). Longitudinal muscle bundles (20–26 LMB, $n = 3$).

**Remarks.** The case of *Aspidosiphon* (*Paraspidosiphon*) *ochrus* is complex. *Cutler & Cutler (1979)* described this species based on specimens from Madagascar (holotype), Papua New Guinea (paratype) and Cocos-Keling Islands (additional material), noting that it differs from *A.* (*P.*) *steenstrupii* by having a lighter anal shield, more longitudinal muscle bundles, and the nephridia attached over a greater extent of the body wall. However, *Cutler & Cutler (1989)* later synonymized their own species with *A.* (*P.*) *steenstrupii*, stating that this species includes worms with anal shield with a range of colors from white to dark brown. They observed that "*Atlantic Ocean populations have dark anal shields, the mid-Pacific Ocean populations have pale shields, and the Indian Ocean population exhibit a*

*mixture, with a higher frequency of dark shields in populations near continent, rare in island populations*".

We were unable to find the holotype, which is apparently deposited at the Muséum National d'Histoire Naturelle, Paris, but we did find the paratype (Figs. 4A–4H) from east of Bougainville Island, Papua New Guinea. Due to a lack of authorization, we did not dissect the paratype, but we supplemented our redescription with the specimens from the Cocos-Keling Islands.

After examining specimens of *A. (P.) steenstrupii* and *A. (P.) ochrus* and reviewing the drawings and description by *Cutler & Cutler (1979)*, we propose that *A. (P.) ochrus* be reinstated. This proposal is based on critical morphological differences between both species. First, we have not found any specimens from the Greater Caribbean with pale anal shields like those of *A. (P.) ochrus*. Second, specimens from the Greater Caribbean with calcareous material on the anal shield retain a crater-like form in lateral view. In contrast, the anal shield of *A. (P.) ochrus* is flat in lateral view, even when covered with calcareous material. Notably, the shape of the shield remains consistent despite the presence of calcareous material. Additionally, we did not find the tongue-like extension in any hooks of *A. (P.) ochrus*, a diagnostic characteristic in *A. (P.) steenstrupii*.

These significant morphological differences between two geographically distant species justify reinstating *A. (P.) ochrus* as a distinct species.

### *Aspidosiphon* (*Paraspidosiphon*) *speculator* *Selenka, 1885* reinst. stat.
Figure 6

*Aspidosiphon speculator Selenka, 1885*: 19–20, pl. 4, figs. 24–27.
*Paraspidosiphon speculator.*—*Stephen & Edmonds, 1972*:253–254.

**Type locality.** Saint Vincent, Cape Verde; shallow water.
**Material examined. Syntypes** BMNH.1885.3.27, 1 specimen. BMNH.1885.3.28, two specimens, St. Vincent, Cape Verde, Challenger, Jul 8, 1873.

### Redescription.
Syntype BMNH.1885.3.27.

External morphology. Trunk 15 mm in length (Fig. 6A); rough, brown with opaque body wall; trunk with circular papillae with platelets, different sizes, non-conglomerate (100–300 µm length), arranged scattered throughout the trunk (Figs. 6F–6H).

Introvert retracted, 50% the trunk length. Introvert papillae conical (Fig. 6I), shorter anteriorly, taller posteriorly (25–30 µm tall), arranged in rings, each papilla with denticles. Tentacles not observed.

Introvert with the most anterior hooks (40–45 µm tall, $n = 7$ rings, $n = 52$ hooks) Type A, (compressed, bidentate), arranged in rings (Figs. 6J–6K). Hooks from anterior and posterior rings with the angle between line X and Y more than 90°; clear streak without basal triangle or tongue-like extension. Hooks all Type A with main and secondary tooth,

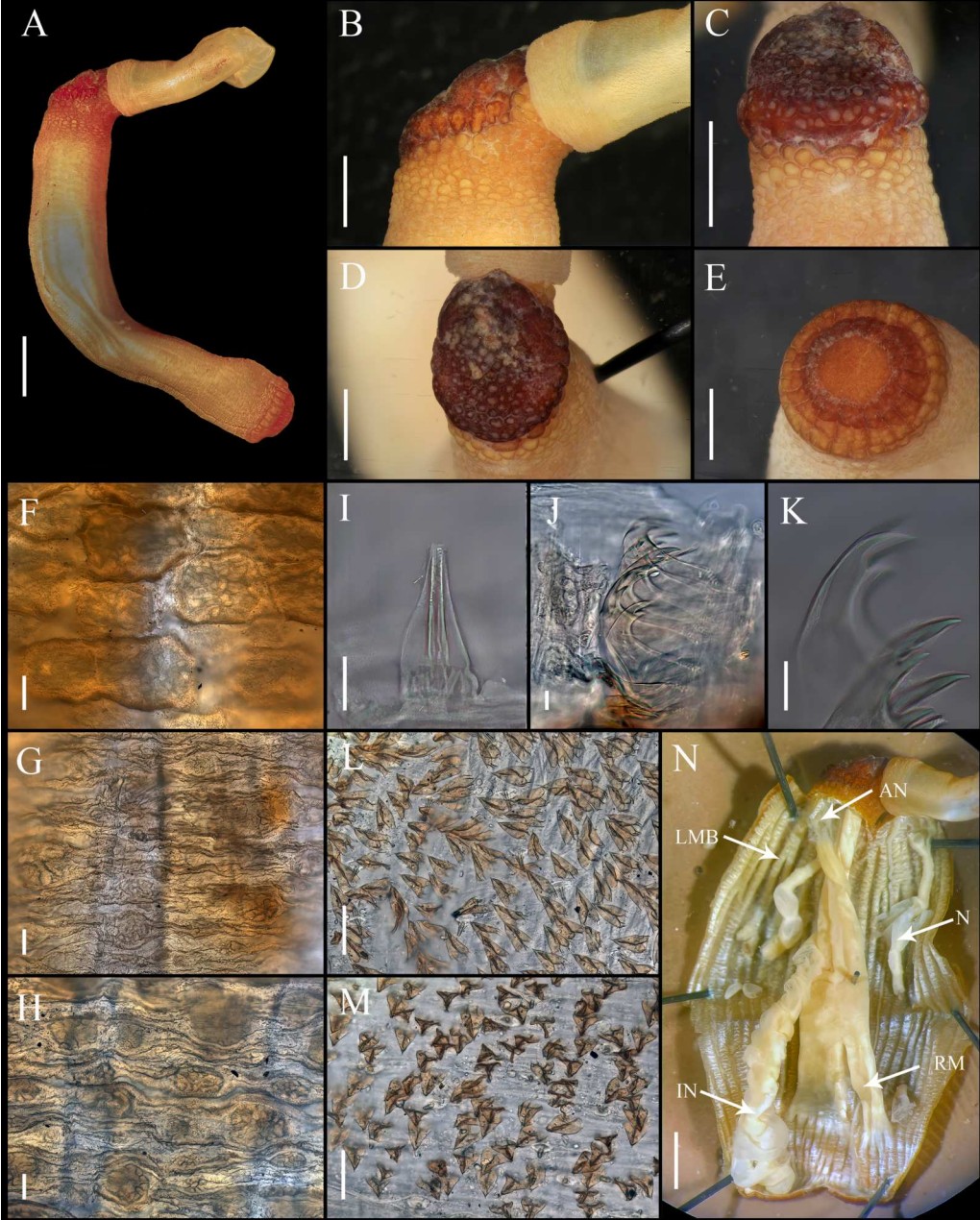

**Figure 6** *Aspidosiphon (Paraspidosiphon) speculator Selenka, 1885* **Syntype BMNH.1885.3.27, Cape Verde.** (A) Adult body plan, lateral view. (B) Anal shield, lateral view. (C–D) Anal shield, dorsal view. (E) Caudal shield, frontal view. (F) Papillae from anterior region of trunk. (G) Papillae from median region of trunk. (H) Papillae from posterior region of trunk. (I) Conical papillae from median introvert. (J) Hooks from anterior rings. (K) Hooks from posterior rings. (L) Leaf-pyramidal hooks from anterior region. (M) Leaf hooks from posterior region. (N) Internal morphology. Abbreviations. AN, anus; IN, intestine; LMB, longitudinal musculature bundles; N, nephridia; RM, retractor muscles. Scale bars. A–E, N: one mm, F–H:100 μm, I–K: 10 μm, L–M: 50 μm.

sharp, main tooth do not exceed the length of the hook base. Hooks Type A followed by Type D, dark leaf hooks (40–50 μm tall), arranged scattered (Figs. 6L–6M).

Anal shield dark brown, thick, margins ill-defined without grooves (Figs. 6B–6D). Conglomerate spherical units in the margins. Shield with inconspicuous crater-like form with an ill-defined conical protuberance.

Caudal shield dark brown with shallow grooves arranged radially, margins ill-defined, semi-rectangular units in the anterior margin (Fig. 6E).

Internal morphology (Fig. 6N). A pair of nephridia opening at the anus level, occupying 55% trunk length. Longitudinal muscle layer divided into 26 anastomosed LMB in the median region of the trunk, anastomosed. A pair of retractor muscles attached to the body wall in the 80% of the trunk length, each retractor muscle attached to 8 bundles, starting from the second bundle after the ventral nerve cord, fused on the half of their length. Fixing muscle present. Caecum and eyespots not observed. Spindle muscle attached posteriorly.

**Variations.** Trunk length (12–15 mm, $n = 3$). Nephridia length (50–55% trunk length, $n = 3$). Retractor muscles attachment (75–80% trunk length, $n = 3$). Longitudinal muscles bundles (25–26 LMB).

**Habitat.** Shallow water.

**Distribution.** *Aspidosiphon* (*P.*) *speculator* is only known from the type locality.

**Remarks.** *Selenka (1885)* described this species after examining three specimens from St. Vincent (Cape Verde Islands), the longest of which had a trunk length of 14 mm.

The differences between the original description and our observations are as follows: He counted 22 LBM ''anastomosing in a complex manner'', while we counted 26 in the median region of the trunk. The high degree of anastomosis explains why the number of bundles may vary depending on the observer.

Selenka did not provide a differential diagnosis or key to explain how his species differs from others previously described within the genus *Aspidosiphon*. *Cutler & Cutler (1989)* noted this absence and highlighted that in *Stephen & Edmonds (1972)*, A. (*P.*) *speculator* and *A.* (*P.*) *steenstrupii* are recognized based on the attachment of the retractor muscles, which falls within the range of *A.* (*P.*) *steenstrupii*. This led them to consider *A.* (*P.*) *speculator* as a junior synonym of *A.* (*P.*) *steenstrupii*.

However, after examining both species, we found substantial morphological differences. The two most notable differences are the inconspicuous anal shield of *A.* (*P.*) *speculator* compared to the conspicuous crater-like shape of *A.* (*P.*) *steenstrupii*. Additionally, the bidentate hooks of *A.* (*P.*) *steenstrupii* exhibit a remarkable tongue-like extension which is absent in *A.* (*P.*) *speculator*. In the more 50 hooks we reviewed from the specimens, and as previously illustrated by Selenka, this feature is clearly lacking. Based on this evidence, we propose the reinstatement of *A.* (*P.*) *speculator* as a distinct species.

## Molecular analysis

The molecular analysis based on nucleotide sequences of the cytochrome c oxidase subunit 1 (COI) gene corroborates the morphological data shown here to recognize the species (Fig. 7). The sequences from *A.* (*P.*) *steenstrupii* from the Greater Caribbean (Florida, Barbados, and Panama) were grouped with an intraspecific variation of 1.99%.

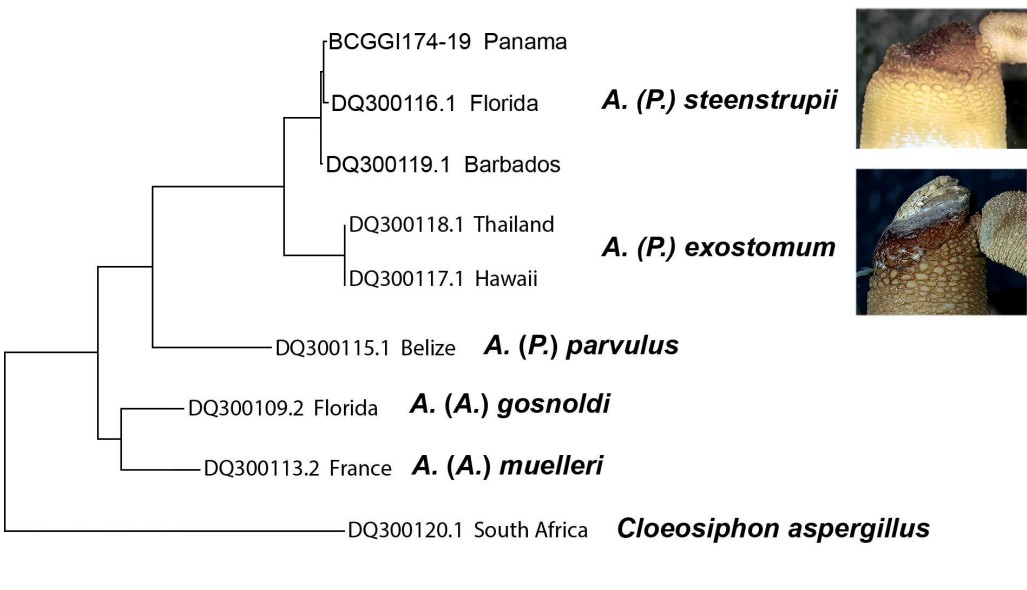

BCGGI174-19 Panama

DQ300116.1 Florida — *A. (P.) steenstrupii*

DQ300119.1 Barbados

DQ300118.1 Thailand — *A. (P.) exostomum*

DQ300117.1 Hawaii

DQ300115.1 Belize — **A. (P.) parvulus**

DQ300109.2 Florida — **A. (A.) gosnoldi**

DQ300113.2 France — **A. (A.) muelleri**

DQ300120.1 South Africa — **Cloeosiphon aspergillus**

0.20

**Figure 7** **Maximum likelihood tree of COI sequences.** Using Tamura-Nei 1993 model with a discrete Gamma distribution with five rate categories and by assuming that a certain fraction of sites is evolutionarily invariable (TN93+G +I).

In contrast, the sequences from *A. (P.) exostomum* from the Pacific (Thailand and Hawaii) showed no intraspecific variation but with an interspecific difference of 19.3% regarding *A. (P.) steenstrupii*. Incidentally, we examined morphologically two specimens of both *A. (P.) steenstrupii* and *A. (P.) exostomum*, which were used to obtain the COI sequences to confirm the accurate species identification.

The result of genetic analysis supports our proposal to reinstate *A. (P.) exostomum* as a distinct species, rejecting the synonymy proposal suggested by *Cutler & Cutler (1989)*. Furthermore, it provides evidence that the distribution of *A. (P.) steenstrupii* is restricted to tropical western Atlantic, contrary to the idea of having a wide distribution throughout the world.

## DISCUSSION

We have divided the discussion of this work into two main sections. First, we will discuss the diagnostic characters valid for recognizing species of aspidosiphonids based on the characters reviewed by *Cutler & Cutler (1989)* and those used thus far to identify species (*Cutler, 1994*), including our comments and proposals. Second, we will address the topic of synonyms of *A. (P.) steenstrupii*.

### Morphology

*Cutler & Cutler (1989)*, based on their morphological observations, concluded the following: "*Hook and anal shield morphology are determined to be broadly useful at the species level, four characters (longitudinal muscle layer, retractor muscles origins, caudal*

*shield, nephridia length) in a more restricted manner to separate subgroups, and three (introvert/trunk angle, bifurcated anterior spindle muscle, loosely wound gut coil) are useful in special cases*". In their revision, the authors discussed the following 11 characters: (1) Introvert hooks, (2) Anal shield, (3) Caudal shield, (4) Introvert retractor muscles, (5) Spindle muscle, (6) Fixing muscle, (7) Nephridia, (8) Rectal caecum, (9) Intestinal coils, (10) Longitudinal muscle, (11) Angle of introvert to trunk.

After our morphological review, based on examining over 100 specimens from a geographically restricted populations, we concur with some of the conclusions of *Cutler & Cutler (1989)*, but we propose some specific modifications. Below, we discuss each character previously discussed in the global review of aspidosiphonids by *Cutler & Cutler (1989)*.

**1. Introvert hooks.** We agree that hooks are one of the most useful morphological characters for recognizing species of aspidosiphonids. For example, several authors have corroborated that species such as *Aspidosiphon (Akrikos) albus* Murina, 1967 lack hooks. On the other hand, some species may have unidentate hooks scattered in the most proximal portion of the introvert, or, as in most species, there may be hooks arranged in rings (either unidentate or bidentate) followed by hooks arranged scattered.

*Cutler & Cutler (1989)* defined three types of hooks. After obtaining scanning electron microscope photographs, we found a substantial morphological difference between the typical pyramidal hooks of species such as *Aspidosiphon (P.) fischeri* and *A. (P.) parvulus* compared to those of *A. (P.) steenstrupii*. Therefore, we propose using a new term to refer to these hooks. Following the terminology of *Cutler & Cutler (1989)*, we suggest defining these as Type D: Leaf Hooks, due to the leaf-like shape of these structures (see 'Materials & Methods').

We disagree with the statement that the clear area in the hooks "has limited taxonomic value" (*Cutler & Cutler, 1989*). Our examination revealed that this character is consistent within a population but varies when compared with other geographically distant populations. We recommend using this feature but describing the variation throughout the introvert.

We also propose including the angle between the main tooth and the body of the hook in the hook's description and thoroughly reviewing the most proximal, distant, and median regions of both the ringed and dispersed hooks.

**2. Anal shield.** We agree that the degree of development of the anal shield is a helpful characteristic for recognizing species. For example, adults of *Aspidosiphon (Akrikos) mexicanus* (Murina, 1967) or *A. (Akrikos) thomassini Cutler & Cutler, 1979* have a poorly defined shield, appearing as a collection of small, scattered units resembling an area of rough skin, compared to species like *A. (Aspidosiphon) muelleri* Diesing, 1851 and *A. (P.) laevis*, where the units are more compact, forming a solid mass (*Cutler & Cutler, 1989*). The authors considered the presence of grooves as a morphological characteristic that can differentiate species, noting that grooves can be longitudinal, transverse, or even absent. Based on our observations, we concur that the anal shield, much like the hooks, is one of the most important morphological characters for species recognition, but we consider the following additional aspects.

It is crucial to describe in detail the units surrounding the anal shield, consider the percentage of groove coverage, and determine whether the grooves are deep or shallow. Additionally, describing the lateral view of the shield is essential, mainly when calcareous material is present. We have observed that the lateral shape remains consistent even with calcareous material. For example, the flat shield of *A.* (*P.*) *ochrus*, the pineapple-shaped shield of *Cloeosiphon aspergillum* (*De Quatrefages, 1866*), and the crater-shaped shield of *A.* (*P.*) *steenstrupii* all maintain their forms despite calcified deposits.

The shield color should not be considered a definitive characteristic, as we have observed that it can range from light brown to nearly black within the same population. *Cutler & Cutler (1989)* noted a geographical variation, with Atlantic Ocean populations having dark shields, mid-Pacific Ocean populations being pale, and Indian Ocean populations exhibiting a mixture (a higher frequency of dark shields in populations near continents, but rare in island populations).

Finally, caution is needed when defining the color of anal shields, as dark shields can appear "pale" if calcareous material is not removed. For instance, the "pale" shield of *A. (P.) ochrus* might be masking the actual color of the anal shield in that population. This also serves as evidence that even with deposited material, the flat shape of this species is maintained. Similarly, in the case of *A. (P.) steenstrupii*, the crater shape is preserved regardless of the presence of calcareous material.

**3. Caudal shield.** According to *Cutler & Cutler (1989)*, the shape of the caudal shield can vary significantly between live specimens and those that are fixed. We agree that this characteristic varies within the same population, regarding its grooves or degree of development. Therefore, aside from its presence or absence (*e.g.*, *A. (Akrikos) mexicanus* and *A. (Akrikos) zinni* Cutler, 1969 have a very inconspicuous caudal shield, whereas *A. (P.) laevis* always has a conspicuous anal shield), the caudal shield offers limited taxonomic.

**4. Introvert retractor muscles.** These muscles are attached to the body wall and participate in the retraction movement of the introvert. In the original descriptions of aspidosiphonids, there has been a lack of precision in describing the position where they are inserted into the body wall. We agree that this feature must be used cautiously, as there is a range of variation in populations (*e.g.*, *A. (P.) steenstrupii* 77–82% trunk length), and there are species in which these muscles insert at the caudal region (100% trunk length). It is important to emphasize that when referencing trunk length, we refer to the distance between the anus and the extreme of the caudal region. Additionally, we concur that the degree of fusion of the retractor muscles is not helpful as a diagnostic character, as this degree of fusion depends on the introvert extension and the developmental stage of the organism (*Rice, 1976*). Lastly, the number of bundles the retractor muscles attach appears consistent among the species reviewed here.

**5. Spindle muscle.** This feature is difficult to observe, even when stained with Shirlastain. Previous observations indicate that the bifurcation of this muscle can only be seen in species such as *A. (P.) laevis* and *A. (P.) coyi*. However, variations in this characteristic have not been thoroughly examined across geographically distant localities. What is certain is that the spindle muscle inserts posteriorly in all species of the family.

**6. Fixing muscle.** Regarding this characteristic, we agree with *Cutler & Cutler (1989)* on the unsuitability of this feature to recognize species because its description is highly dependent on the dissection quality. Nevertheless, most importantly, we observed that it is a morphological character that may or may not be present within the same geographically restricted population.

**7. Nephridia.** Nephridia are considered diagnostic characters in some cases. For instance, their length, the percentage of attachment to the body wall, and the positioning of the nephridiopore relative to the anus are regarded as helpful in differentiating species. After our morphological review, we can conclude that, in the case of *A. (P.) steenstrupii* and the species examined in this study, the length of the nephridia within a single population can vary from 55–82% of the trunk length. This range of variation overlaps with the populations that have been reviewed so far. During dissection, the ligaments connecting the nephridia to the body wall are susceptible to detachment, potentially causing discrepancies between the observed condition and the typical state of this feature. Therefore, we do not regard the percentage of nephridia attachment as a reliable characteristic for species differentiation.

**8–9. Rectal caecum and intestinal coils.** We consider the rectal caecum an unreliable characteristic for species differentiation, as our observations reveal that it is present in only some individuals within the same population. Regarding intestinal coils, we have noted that within a single population, the number of coils can vary as the torsion of the intestine is affected by the contraction state of the organism. Like the rectal caecum, we find this character unhelpful for species recognition. Furthermore, locating the caecum and counting the intestinal coils is quite challenging, making it difficult to describe as a character specific to any population. It also depends on the observer's interpretation.

**10. Longitudinal muscle.** Aspidosiphonids can have bundles of longitudinal musculature along the entire trunk. However, some species show no signs of such bundles, suggesting that the longitudinal musculature consists of a single continuous layer. A third scenario includes some divisions beneath the anal shield. To standardize this situation, we propose considering three-character states based on the percentage these bundles cover: (1) Less than 50% of the trunk length, (2) More than 50%, and (3) No divisions. In the case of *A. (P.) steenstrupii* and the species we reviewed; the bundles extend along the entire trunk. The bundles in the species we examined are anastomosed, meaning that there are various connection points, unlike, for example, the notorious separated bands of the Sipunculidae family. This degree of anastomosis can confuse determining the number of bundles in a specimen.

**11. Angle of introvert to trunk.** *Cutler & Cutler (1989)* described this characteristic regarding the angle between the main axis of the trunk and the ventral side of the anal shield. It is true that some species have an angle ranging from 45–60°, while in others it ranges from 75–90%. This characteristic is closely related to the degree of development of the anal shield. Typically, species with less defined anal shields tend to have angles smaller than those found in most aspidosiphonid species.

## Synonyms

*Cutler & Cutler (1989)* conducted an analysis using several specimens from various localities to describe the morphological variation. Although the worldwide revision attempted to standardize descriptions, this goal was not fully achieved. Many descriptions were poorly elaborated, and some were based on specimens from different and very distant localities from the type localities. They observed living material and discussed those taxa not belonging to *Aspidosiphon*. This was a significant contribution; however, with our work, the substantial contributions of both authors have been enhanced in the following way. Our observations are based on the description of geographically distant populations independently, an essential consideration because *Cutler & Cutler*'s (*1989*) descriptions were based on a mix of specimens from geographically distant localities. Moreover, considering that Sipuncula is a group with very few diagnostic characters, we consider that both authors might have underestimated the morphological differences they observed, attributing them to a variation among widely distributed species.

*Cutler & Cutler (1989)* included eight species as junior synonyms of *A.* (*P.*) *steenstrupii*: *A. semperi* from Caracas Bay, *A. makoensis* from Mako, Formosa, Taiwan, *A. trinidensis* from Trinidad Island, Brazil, *A. exostomum* from Andaman Islands, *A. speculator* from Cape Verde, *A.* (*P.*) *ochrus* from Madagascar, *A. fuscus Sluiter, 1881* from Malay Archipelago and *A. steenstrupii* var. *fasciatus* (*Augener, 1903*) from Ambon Island.

As noted above, we restrict the distribution of *A.* (*P.*) *steenstrupii* to the tropical Western Atlantic. The morphological difference used to recognize the species *A. semperi* (from Curaçao) is questionable. Although we could not review the type material for this species, the most likely hypothesis is that the presence of four retractor muscles was an observational errors because all aspidosiphonid species have always two retractor muscles. Regarding *A. makoensis*, *Cutler & Cutler (1981)* synonymized it with *A.* (*P.*) *steenstrupii*; we have discussed the hooks as an essential diagnostic character; therefore, the differences between the hooks of *A. makoensis* and *A.* (*P.*) *steenstrupii* are sufficient to propose its reinstatement, pending the review of topotype material to confirm it.

In the case of *A.* (*P.*) *exostomum*, *A.* (*P.*) *ochrus*, and *A.* (*P.*) *speculator*, we were able to review type material, which confirmed morphological differences in the hooks and the anal shield that allowed us to argue for the validity of these three species names. The species *A. fuscus* and *A. steenstrupii* var. *fuscus* could not be located, but searching for topotypic material to clarify their taxonomic status is recommended.

After this revision, we evaluated the validity status of some synonyms. With our proposition for reinstating three species, we contribute to rejecting the previous hypothesis of species with a cosmopolitan distribution. Recently, this idea has been supported by some studies based on detailed morphological revision and molecular data, showing evidence that slight morphological differences represent different species (*Staton & Rice, 1999*; *Schulze et al., 2012*; *Kawauchi & Giribet, 2010*; *Kawauchi & Giribet, 2014*; *Johnson et al., 2016*; *Silva-Morales et al., 2019*; *Silva-Morales, 2020*).

## CONCLUSION

We delimited *A.* (*P.*) *steenstrupii* morphologically, restricting its distribution to the tropical Western Atlantic. We propose that the anal shield and the variation of hooks along the introvert are essential for describing aspidosiphonid species, as these features allowed us to propose the reinstatement of three previously synonymized species. The molecular analysis confirmed our observations.

With the findings from various studies and our results, we consider that the diversity of sipunculans is underestimated. We encourage detailed morphological studies species-by-species on sipunculans to evaluate the status of synonyms and contribute with complete redescriptions using a combination of current tools. This approach will help to determine a more accurate count of extant sipunculan species worldwide.

## ACKNOWLEDGEMENTS

We thank Gustav Paulay (UF) for the loan of materials and the provided nucleotide sequence. We also thank Nancy Voss and María Criales (UMML) for making available some of the materials that made this study possible. We thank Adam Baldinger (MCZ), Birger Neuhaus (MFN), Karen Osborn and Karen Reed (USNM) for their kindly lending the materials and managing materials requests during Itzahí stays at MCZ, MFN, and USNM. Thanks to Fiona Ware (NMS), Emma Sherlock (BMNH), and Tarik Meziane (MNHN) for agreeing to send materials from your collections to Berlin. Leticia Franco and three anonymous reviewers helped us a lot to improve the manuscript.

### Funding

This work was supported by Institutional funds of Luis F. Carrera-Parra. During this research, Itzahí Silva-Morales was supported by a scholarship from CONAHCYT and the following scholarships were obtained for stays in museums: Ernest Mayr Grant, Museum of Comparative Zoology, Harvard University; Kenneth Jay Boss Fellowship, National Museum of Natural History, Smithsonian Institution; Short-term grant 2023 Deutscher Akademischer Austauschdient, Berlin. There was no additional external funding received for this study. The funders had no role in study design, data collection and analysis, decision to publish, or preparation of the manuscript.

### Grant Disclosures

The following grant information was disclosed by the authors:
Institutional funds of Luis F. Carrera-Parra.
CONAHCYT.
Ernest Mayr Grant, Museum of Comparative Zoology, Harvard University.
Kenneth Jay Boss Fellowship, National Museum of Natural History, Smithsonian Institution.
Deutscher Akademischer Austauschdient, Berlin.

## Competing Interests

The authors declare there are no competing interests.

## Author Contributions

- Itzahi Silva-Morales conceived and designed the experiments, performed the experiments, analyzed the data, prepared figures and/or tables, authored or reviewed drafts of the article, and approved the final draft.
- Luis F. Carrera-Parra conceived and designed the experiments, performed the experiments, analyzed the data, prepared figures and/or tables, authored or reviewed drafts of the article, and approved the final draft.

## DNA Deposition

The following information was supplied regarding the deposition of DNA sequences:

The sequences are available at GenBank: *Aspidosiphon (P.) steenstrupii* (DQ300119.1, DQ300116.1, *Aspidosiphon steenstrupii* voucher UF495 (BCGG|174-19) cytochrome c oxidase subunit I (COX1) gene, partial cds: PV036402), *A. (P.) exostomum* (as *A. (P.) steenstrupii* DQ300118.1, DQ300117.1), *A. (P.) parvulus* (DQ300115.1), *A. (A.) gosnoldi* (DQ300109.2), *A. (A.) muelleri* (DQ300113.2), *Cloeosiphon aspergillus* (DQ300120.1).

## Data Availability

The specimens reviewed of *Aspidosiphon (P.) steenstrupii* are deposited in the Marine Invertebrate Museum (UMML), Rosenstiel School of Marine and Atmospheric Science, University of Miami, Florida, USA. Museum of Comparative Zoology (MCZ), Harvard University, Massachusetts, USA. Invertebrate Collections of the Florida Museum of Natural History (UF), University of Florida, USA. Colección de Bentos Costero (ECOSUR), El Colegio de la Frontera Sur, Chetumal, Mexico.

The specimens of *Aspidosiphon (P.) exostomum* are deposited in the National Museums of Scotland (NMS), Edinburgh, Scotland and the Museum of Comparative Zoology (MCZ), Harvard University, Massachusetts, USA.

The specimens of *Aspidosiphon (P.) ochrus* are deposited in the Muséum National d'Histoire Naturelle (MNHN), Paris, France and the National Museum of the Natural History (USNM), Smithsonian Institution, Washington, USA.

The specimens of *Aspidosiphon (P.) speculator* are deposited in The British Museum of Natural History (BMNH), London, England.

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
