# Peer review of "Redescription of Aspidosiphon (Paraspidosiphon) steenstrupii Diesing, 1859 (Sipuncula: Aspidosiphonidae) and the reinstatement of three species"

_PeerJ, doi:10.7717/peerj.19003_

## Round 0.1 · original submission · Major Revisions

Dear authors, I ask you to respond very carefully to all the fundamental comments on your manuscript. I hope that the new version of this manuscript you have sent will allow the reviewers to approve the publication of this article.

Reviewer 1 ·

Basic reporting

Overall, the paper is well written but at times quite wordy. I have inserted multiple comments and edits in the enclosed pdf.

Experimental design

No issues

Validity of the findings

Valid findings

Additional comments

The manuscript is a valuable contribution to clarifying the taxonomy of an understudied group of marine worms. Their detailed morphological examinations have revealed diagnosable differences between Aspidosiphon (Paraspidosiphon) steenstrupii and the three previously synonymized species they are reinstating. Based on multiple previous molecular analyses, which have shown that many sipunculan species are actually complexes of cryptic or pseudo-cryptic species, the findings are not surprising. Taxonomy is a challenging and time-consuming task which does not receive much funding. It is therefore very laudable that the authors tracked down as much type material as possible and visited various collections to examine the material. The merit of the present manuscript lies in the careful documentation of the morphological characters, excellent imagery and careful historical research.
The molecular analyses are only of value with regard to one of the reinstated species - A. (P.) exostomum – because no sequence data are available for A. (P.) ochrus or A (P.) speculator. The statement “Furthermore, we discuss the taxonomic status of synonyms based on morphological and molecular data” (end of Introduction) is therefore misleading, as it implies that an integrative taxonomy approach was used for all the re-instated synonyms.

Annotated reviews are not available for download in order to protect the identity of reviewers who chose to remain anonymous.

·

Basic reporting

The article is well-written and well-supported by the literature. However, there are a few areas that need improvement. In the abstract, line 19, "Western America" should be changed to "Western Atlantic," as the authors are referring to marine animals. Also, in line 23, "compreensive" should be corrected to "comprehensive." In the introduction, I suggest improvements in English, such as replacing "have" with "face" (line 31), "perpective" with the plural form (line 33), and "recognation" with "identification" (line 36). Additionally, "Aspidosiphon" is misspelled in line 45.

In the Results section, I recommend changing "height" to "tall," as it is the traditional way to refer to hook height according to Cutler & Cutler (1989), and the term “dispersedly” should be avoided due to its ambiguity regarding whether it means "spaced out" or "in a disorganized manner."

There are also confusing sentences that could be improved, particularly lines 35-38, where the lack of taxonomic characters is attributed to brief original descriptions and the absence of illustrations. However, I believe this should not be seen as a direct cause-and-effect situation. It is important to reference Schulze et al. (2012) and Saiz-Salinas (2018) to support this argument. These works discuss cosmopolitan species in Sipuncula and the lack of reliable diagnostic characters in the group, which are significant factors in the difficulty of species differentiation. Additionally, in line 128, Sipuncula is incorrectly referred to as a phylum. Recent molecular analyses suggest that Sipuncula is a clade within Annelida (Weigert & Bleidorn, 2016).

Another point to address is the mention of hook counts in lines 234-235. The number of rings should be introduced after the mention of the hooks being arranged in rings, which would improve clarity. Additionally, in line 240, a new type of hook is introduced without proper explanation or reference. Cutler & Cutler (1989) describe three types of hooks for Aspidosiphon and mention that Aspidosiphon (Paraspidosiphon) steenstrupii has Type A and B hooks. It is essential to discuss why the classification has changed.

Finally, the species redescriptions lack a diagnosis section, which is crucial for clearly differentiating the re-established species from congeners. Including this section would be valuable for the taxonomy and essential in any taxonomic work.

Concerning figures and tables, most images are useful and contribute significantly to illustrating the redescribed species. However, some improvements are needed. In Figure 1J-K, the hook photos appear slightly blurry. I suggest either redrawing them or editing the photos for better clarity. In Figure 3R, the internal structures are hard to distinguish due to the background clutter and the visible shadow from the ocular lens. I recommend cleaning up the background or providing a detailed drawing of the internal anatomy of Aspidosiphon (Paraspidosiphon) exostomum. Another point is that in the legend of Figure 3, the reference to Figure 3R is missing. Including it would ensure completeness and clarity in the figure description. Additionally, the article would benefit from a table listing genetic distances, sequences generated, specimen localities, and accession numbers.

Experimental design

Undoubtedly, the topic of this study is highly significant within the systematics of Sipuncula. Revising species with extensive synonymy lists is essential for a better understanding of the group's diversity. This is a well-executed study with thorough redescriptions and a commendable effort by the authors to revisit type specimens. Moreover, the authors' expertise in dissecting such small specimens, like Aspidosiphon, is evident and impressive.

Regarding the methodology, my only concern is in the Materials and Methods section (lines 110-114), where molecular sequence additions are mentioned without detailing how they were obtained, including the primers used or the extraction and amplification protocols. Including these details is critical to ensure the reproducibility of the study.

Another point is about the type specie of Aspidosiphon (Paraspidosiphon) steenstrupii, on line 227 the authors redescribe the specie base on a Male specimen from Saint Martin, West Indies (UF331), on the remarks they discuss that is not the holotype and is an specimen for a close locality from the type, but they not explain where is the holotype. I imagine that the type species is lost, but if is the case i recommend they establish a neotype or topotype

On the use of middle counts of longitudinal muscle bands, Cutler & Cutler (1989) report only anterior and posterior counts. While the authors argue (lines 395–396) that middle counts are better due to anastomoses, no data is provided to support lower variation in this region and why this count is better.

Finally, when reporting morphological variation (lines 259–262) and on the other variation topis, specifying the number of specimens measured for each character would enhance accuracy and clarify the sampled variation's scope. Addressing these points would improve the study's methodological transparency and robustness.

Validity of the findings

Strengths:
- The article demonstrates the authors’ meticulous effort in species descriptions and examinations.
- The study provides valid and consistent evidence for reinstating A. (P.) exostomum, A. (P.) ochrus, and A. (P.) speculator.
- Additional specimens, beyond the type material, enrich the descriptions.
Incorporating molecular analyses, even if only to differentiate A. (P.) steenstrupii from A. (P.) exostomum, adds value. Although sequences of the other reinstated species were not analyzed, the study remains robust, providing morphological characters that distinguish the four species.
- The arguments and ideas are well-supported, showcasing mastery of the historical literature on the group.
Weaknesses:
Detailed Descriptions:
- While dissecting small Sipuncula specimens is challenging, the descriptions could be more detailed:
- The introvert is stated to be shorter than the trunk, but no measurements or proportions are provided. These would enhance the redescriptions.
- Hook tall for A. (P.) steenstrupii, A. (P.) exostomum, and A. (P.) speculator are not provided, limiting comparisons with other species.
- Only the median trunk region is used for counting longitudinal muscle bands, whereas Cutler & Cutler (1989) included anterior and posterior counts. Adding these would improve the manuscript.
Additional Characters:
- The position of nephridia openings relative to muscle bands is not mentioned.
- The proportion of the nephridium attached to the body is discussed but lacks specific data from the specimens analyzed.
- The presence or absence of wing muscles is omitted.
- Fixing muscle positions and spindle muscle anterior attachments are not included. If observations were not possible, this limitation should be acknowledged.
Comparative Issues:
- Lines 394–396 compare results with Johnson (1964), but the latter counted muscle bands in anterior/posterior regions, making the comparison incongruent.
- The significant difference in the number of hook rings (20 in A. (P.) steenstrupii vs. 7–8 in others) is notable but unaddressed.
Questionable Points:
- Type D Hooks (Lines 621–628): The distinction between Type D hooks and the pyramidal hooks described by Cutler & Cutler (1989) requires clarification. Figure 1G of Cutler & Cutler (1989) depicts a Type B hook, which appears similar to the authors’ Type D. While recategorization may be valid, the proposed new type lacks convincing differentiation.
- Dissection and Nephridium (Lines 696–698):The phrasing suggests nephridium attachment varies with dissection quality. This is misleading since attachment is a natural trait. Instead, clarify that dissection affects visibility, and ambiguous terms like "poor relaxation" should specify improper preparation techniques.
- Cecum Taxonomy Utility (Lines 699–701): While the authors argue the cecum is not useful for species differentiation, this assertion requires examination of the structure. Without thorough observation, its relevance cannot be dismissed.

Additional comments

The article under review is outstanding and represents a significant advancement in Sipuncula systematics and taxonomy. Experts in Sipuncula are few, and even fewer are willing to address species as small as those in the genus Aspidosiphon. I admire the authors' commitment to contributing such meaningful discoveries. A meticulous, exhaustive, and significant work like this is exactly what the field of Sipuncula systematics needs. I reviewed the paper with great pleasure, and I believe all my suggestions and questions will further enhance this excellent work.

Reviewer 3 ·

Basic reporting

This manuscript can be published after some changes have been made. The authors checked many specimens from different collectons. The main statement of the manuscript that the authors reported that their main task is to create a uniform description of sipunculans.
The restatement of Aspidosiphon speculator looks too speculative. I would insist on removing of the Identification the species returning to valid species list. The author may leave this to the discussion, but it is not enough data to judge the taxonomic status of species. It has similar external and internal characteristics, and the anal shield looks identical (Same hooks size, same LMs number, same attachment of RMs…).
The validation of the existence of Aspidosiphon ochrus is much obvious. But specimens from Papua New Guinea and the Indian Ocean do not look too similar, even the caudal shields of both have whitish calcareous units. Regarding the color of the shields of the Aspidosiphons, it was mentioned in Adrianov and Maiorova, 2012 that in different habitates shiels color may vary “Anal shield dark colored (brown or black) in specimens from gastropod shells and pale-yellow in specimens from dead corals”.
Within the Sipuncula, in many cases, the rapid restatement of species will bring us back to the same point where Cutler and Cutler started.
Line 31 ” Accordig to WORMs this number is already changed, with the effort of the authors of present manuscript as well. It may be better to use and independently updated accounting service, instead of articles.

Experimental design

no comments

Validity of the findings

the data presented in the manuscript helps to correct sipunculan identifications in future

Additional comments

no comments

Reviewer 4 ·

Basic reporting

The work is clear and presents relevant results to the broad understanding of the diversification of an important group of marine invertebrates, The work is generally well-written with some needed modifications to improve the descriptions of the species.

Experimental design

The methods described have sufficient detail and information to replicate.

Validity of the findings

The findings are well substantiated and with implications to the taxonomy of Sipuncula, an important group within Annelida. The morphological descriptions are well done and will set the tone for future taxonomic work in Aspidosiphonidae. The conclusions are well stated but there are some required information about the type material os A. steenstrupii that need to be more explicit.

Additional comments

I have made suggestions within the attached PDF.

Annotated reviews are not available for download in order to protect the identity of reviewers who chose to remain anonymous.

---

## Round 0.2 · Minor Revisions

Dear authors, I ask you to carefully correct the final shortcomings of the manuscript before it is accepted for publication.

Reviewer 1 ·

Basic reporting

As previously, I congratulate the authors on their careful taxonomic work. The figures are high quality and illustrate the text well.
In the Introduction, the authors compare Sipuncula to polychaetes which is a bit controversial because most recent phylogenomic analyses place the Sipuncula as an early branch within the annelid radiation. Although this debate is beyond the scope of the paper, I think it should briefly be mentioned.
Overall, the manuscript is still way too long. It could probably be reduced to about 3/4 of its current length. I have inserted a number of edits and comments to this regard into the manuscript.

Experimental design

No comments

Validity of the findings

The observations are well documented and support the conclusions.

Additional comments

There are several instances where the text is repetitive. For example, the description of the new hook type is almost the same in the Methods section and in the Discussion. Please condense the text.

Annotated reviews are not available for download in order to protect the identity of reviewers who chose to remain anonymous.

·

Basic reporting

The paper was well written and the improvements made it better and clarifier.

Experimental design

It good. The authors incorporate the suggestions.

Validity of the findings

The specification of the hooks types on the methods improves the understanding on the descriptions. I agree with the authors about the decision of not include diagnosis for the descriptions, but to clarify I suggest explain this decision on the discussion. It will be important for futures works and establish an description pattern in Sipuncula.

Reviewer 3 ·

Basic reporting

The authors seriously improved the manuscript. But small bugs are still present.
1. In all descriptions Authors use “smaller ... and longer” to describe papillae, but this pair of words cannot be paired this way. Smaller- larger (or bigger), but not smaller – longer. At least conical papillae cannot be longer, but they can be taller. If taller than shorter. Try to find the pair of words that will help to reader visualize your description.
2. In all species descriptions size of hooks TypeA present only, no size to hooks TypeD present. Please clarify this data, if both hook types are of same size place this also same how.
3. No diagnosis done to any species. Diagnosis must be present when one making description-redescription-reinstatement of new species. According to the present descriptions species cannot be distinguished even between each other.
4. If authors suggested that number of sipunculan species are not correct in WORMs, they may count this number themselves but not use number that other scientist counted four years previously. Since that time some species appeared. Even the authors of this manuscript added few (see(Hsueh and Glasby, 2024; Silva-Morales, 2020; Silva-Morales and Gómez-Vásquez, 2021).
5. Line 350 and 825. Misspelling in the author’s names.
6. Line 352. Aspidosiphon makoensis collected at Mako, that located near Taiwan. Specimens used for the paper Hsueh, Cheng & Kuo, 2006 collected at eastern and most southern parts of Taiwan Island. Specimens from this area need to be twice checked before giving fast decisions.
7. Line 588-594. Images in the manuscript are done with different qualities and positions of view, and they are not very accurate. It is not possible to find differences between them. Reinstatement of the species with this quality of support will bring the value of this reinstatement to the position as Cutler’s synonymy when other scientists cannot identify species correctly, except by geographical distribution. Returning of this species is too speculative at this level of support and specimen illustrations. It is better to leave this data for Discussion or wait until good support for illustrations can be provided. The argument that authors watched the difference between species does not work proper in the taxonomy.
8. Line 633. Better to change “unidentate hooks dispersed” to “unidentate hooks scattered”.

9. Capture to figure 4, correct to Papua New Guinea.

Hsueh, P.-W., Glasby, C.J., 2024. Australian Journal of Zoology. Aust. J. Taxon. 50, 1–5. https://doi.org/https://doi.org/10.54102/ ajt.93q58
Silva-Morales, I., 2020. Reinstatement of Phascolosoma (Phascolosoma) varians Keferstein, 1865 (Sipuncula: Phascolosomatidae) based on morphological and molecular data. PeerJ 8. https://doi.org/10.7717/peerj.10238
Silva-Morales, I., Gómez-Vásquez, J.D., 2021. First records and two new species of sipunculans (Sipuncula) from the Southern Mexican Pacific. Eur. J. Taxon. 740, 77–117. https://doi.org/10.5852/ejt.2021.740.1283

Experimental design

no comment

Validity of the findings

no comment

Additional comments

no comment

---

## Round 0.3 · accepted · Accept

Dear authors, I congratulate you on the acceptance of this article for publication and wish you further success in your scientific work.